



# 1  Experimental chemical budgets of OH, HO₂ and RO₂ radicals in
# 2  rural air in West-Germany during the JULIAC campaign 2019

Changmin Cho[1], Hendrik Fuchs[1], Andreas Hofzumahaus[1], Frank Holland[1], William J. Bloss[3],
Birger Bohn[1], Hans-Peter Dorn[1], Marvin Glowania[1], Thorsten Hohaus[1], Lu Liu[1], Paul S. Monks[2],
Doreen Niether[1], Franz Rohrer[1], Roberto Sommariva[2,3], Zhaofeng Tan[1], Ralf Tillmann[1], Astrid
Kiendler-Scharr[1], Andreas Wahner[1], and Anna Novelli[1]
[1]Forschungszentrum Jülich, Institute of Energy and Climate Research: Troposphere (IEK-8), Jülich,
Germany
[2]Department of Chemistry, University of Leicester, Leicester, UK
[3]School of Geography, Earth and Environmental Sciences, University of Birmingham, Birmingham, UK
*Correspondence to*:  Hendrik Fuchs (h.fuchs@fz-juelich.de) and Anna Novelli (a.novelli@fz-juelich.de)
**Abstract.**
Photochemical processes in ambient air were studied using the atmospheric simulation chamber SAPHIR
at Forschungszentrum Jülich, Germany. Ambient air was continuously drawn into the chamber through a
50 m high inlet line and passed through the chamber for one month in each season throughout 2019. The
residence time of the air inside the chamber was about one hour. As the research center is surrounded by a
mixed deciduous forest and is located close to the city Jülich, the sampled air was influenced by both
anthropogenic and biogenic emissions. Measurements of hydroxyl (OH), hydroperoxyl (HO₂) and organic
peroxy (RO₂) radicals were achieved by a laser-induced fluorescence instrument. The radical measurements
together with measurements of OH reactivity ($k_{OH}$, the inverse of the OH lifetime) and a comprehensive set
of trace gas concentrations and aerosol properties allowed for the investigation of the seasonal and diurnal
variation of radical production and destruction pathways. In spring and summer periods, median OH
concentrations reached $6 \times 10^6$ cm⁻³ at noon, and median concentrations of both, HO₂ and RO₂ radicals,
were $3 \times 10^8$ cm⁻³. The measured OH reactivity was between 4 and 18 s⁻¹ in both seasons. The total reaction
rate of peroxy radicals with NO was found to be consistent with production rates of odd oxygen ($O_X =$
$NO_2+O_3$) determined from $NO_2$ and $O_3$ concentration measurements. The chemical budgets of radicals were
analysed for the spring and summer seasons, when peroxy radical concentrations were above the detection
limit. For most conditions, the concentrations of radicals were mainly sustained by the regeneration of OH
via reactions of HO₂ and RO₂ radicals with nitric oxide (NO). The median diurnal profiles of the total
radical production and destruction rates showed maxima between 3 to 8 ppbv h⁻¹ for OH, HO₂ and RO₂.
Total RO_X (OH, HO₂, and RO₂) initiation and termination rates were below 3 ppbv h⁻¹. The highest OH
radical turnover rate of 13 ppbv h⁻¹ was observed during a high-temperature (max 40°C) period in August.
In this period, the highest HO₂, RO₂ and RO_X turnover rates were around 11, 10 and 4 ppbv h⁻¹, respectively.
When NO mixing ratios were between 1 ppbv to 3 ppbv, OH and HO₂ production and destruction rates
were balanced, but unexplained RO₂ and RO_X production reactions with median rates of 2 ppbv h⁻¹ and 0.4
ppbv h⁻¹, respectively, were required to balance their destruction. For NO mixing ratios above 3 ppbv, the
peroxy radical reaction rates with NO were highly uncertain due to the low peroxy radical concentrations
close to the limit of NO interferences in the HO₂ and RO₂ measurements. For NO mixing ratios below 1
ppbv, a missing OH source with a rate of up to 3.0 ppbv h⁻¹ was found. This missing OH source consists





likely of a combination of a missing primary radical source (0.5 ~ 1.4 ppbv h$^{-1}$) and a missing inter-radical
HO$_2$ to OH conversion reaction with a rate of up to 2.5 ppbv h$^{-1}$. The dataset collected in this campaign
allowed to analyze the potential impact of OH regeneration from RO$_2$ isomerization reactions from isoprene,
HO$_2$ uptake on aerosol, and RO$_2$ production from chlorine chemistry on radical production and destruction
rates. These processes were negligible for the chemical conditions encountered in this study.
**1 Introduction**
The hydroxyl (OH) radical is the dominant daytime atmospheric oxidant. It reacts with most trace gases in
the troposphere and thereby controls the rate of their removal and chemical transformation. In the lower
troposphere, OH is primarily produced by solar photolysis of ozone (O$_3$) and nitrous acid (HONO). The
reaction of OH with trace gases leads to the formation of hydroperoxy (HO$_2$) or organic peroxy (RO$_2$, with
R = organic group) radicals, which undergo further radical reactions. Generally, these reactions are cyclic
chain reactions, in which OH, HO$_2$, and RO$_2$ are converted into each other, while at the same time emitted
pollutants are oxidized and converted into secondary pollutants such ozone and oxygenated volatile organic
compounds (OVOCs). Because the conversion of radicals occurs on a time scale of seconds to minutes,
they are often referred to as the RO$_X$ family (OH + HO$_2$ + RO$_2$). The most important radical reactions in
the lower are summarized in Table 1. Understanding the radical chemistry is the basis for reliable
predictions of the atmospheric lifetime and chemical transformation of air pollutants and climate-relevant
gases by atmospheric chemistry models (Stone et al., 2012).
The level of agreement between simulated and observed radical concentrations in various environments
shows the degree of understanding of the underlying radical chemical mechanism. Even though good
agreement is found in some cases (Tan et al., 2001; Konrad et al., 2003; Mihelcic et al., 2003; Lelieveld et
al., 2008; Kubistin et al., 2010; Whalley et al., 2011), there are significant unexplained discrepancies
between modelled and measured OH in forested regions (Wolfe et al., 2011; Kim et al., 2013; Hens et al.,
2014; Wolfe et al., 2014; Griffith et al., 2016) and of HO$_2$ and RO$_2$ in polluted areas (Ren et al., 2003; Ren
et al., 2006; Kanaya et al., 2007; Dusanter et al., 2009; Chen et al., 2010; Ren et al., 2013; Brune et al.,
2016; Tan et al., 2018; Slater et al., 2020; Whalley et al., 2021), while different results are found depending
on the abundance of nitric oxide (NO) in rural environments (Hofzumahaus et al., 2009; Lou et al., 2010;
Elshorbany et al., 2012; Kanaya et al., 2012; Tan et al., 2017).
A chemical budget analysis using measured OH, HO$_2$ and RO$_2$ radical concentrations can help assessing
the strength of different radical production and loss paths. This allows to identify possible missing chemical
processes by comparing the total production and destruction rates for the different radicals as concentrations
are expected to be in steady-state due to their short chemical lifetime. A large number of measurements
needs to be available (e.g., OH reactivity, OH, peroxy radicals), therefore, there have been only few studies
focusing on the analysis of the chemical budget for OH radicals so far (Handisides et al., 2003;
Hofzumahaus et al., 2009; Brune et al., 2016; Whalley et al., 2018; Tan et al., 2019; Whalley et al., 2021).
Results from field campaigns in China showed a larger OH radical destruction rate compared to its
production rate in the afternoon, which points to an unaccounted OH radical source. Discrepancies were
highest, when NO mixing ratios were lower than 2 ppbv (Hofzumahaus et al., 2009; Tan et al., 2019;
Whalley et al., 2021). On the other hand, studies in urban areas in California (Brune et al., 2016) and in
London (Whalley et al., 2018) as well as in a rural area in Hohenpeissenberg (Handisides et al., 2003)



showed no significant gap between the OH production and destruction rates. Recently, radical
measurements including $RO_2$ enabled the investigation of $HO_2$, $RO_2$, and $RO_X$ production and destruction
rates in field campaigns in China (Tan et al., 2019; Whalley et al., 2021). Tan et al. (2019) showed that a
$RO_2$ loss process was required in a campaign in Wangdu in summer, while $HO_2$ production and destruction
rates were balanced. This suggests a missing conversion of $RO_2$ to OH in addition to the reaction of peroxy
radicals with NO. Furthermore, Whalley et al. (2021) found large imbalances between peroxy radical
production and destruction rates in Beijing indicating a substantially slower propagation of $RO_2$ to $HO_2$
radicals than anticipated.
In this study, OH, $HO_2$, and $RO_2$ radical concentrations as well as OH reactivity, the inverse of the OH
radical lifetime, were measured in the atmospheric simulation chamber SAPHIR on campus of
Forschungszentrum Jülich (FZJ), Germany, in the Jülich Atmospheric Chemistry Project Campaign
(JULIAC). Ambient air was sampled from 50 m height into the SAPHIR chamber. From this data set, a
chemical budget analysis of OH, $HO_2$, $RO_2$ radicals, and their sum ($RO_X$) was done using measured
concentrations allowing to investigate, if all radical production and destruction processes were accounted
for during spring and summer.



**Table 1**. Chemical reactions and rate constants used for the analysis of the chemical budgets of radicals. Values of reaction rate constants are given for standard conditions (298 K, 1 atm). Actual numbers are

| | Reaction | $k$(298 K, 1 atm) / $cm^3\,s^{-1}$ | $k_{ERR}$[a] | Reference |
|---|---|---|---|---|
| **Radical initiation reactions** | | | | |
| R1 | $HONO+h\nu \rightarrow OH + NO$ | $j_{HONO}$[b] | | |
| R2 | $O_3+h\nu \rightarrow O^1D+O_2$ | $j_{O1D}$[b] | | |
| R2a | $O^1D+H_2O \rightarrow 2OH$ | $2.1\times10^{-10}$ | ±13% | IUPAC |
| R2b | $O^1D+M \rightarrow O^3P+M$ | $3.3\times10^{-11}$ | ±10% | IUPAC and JPL |
| R3 | $HCHO+h\nu \rightarrow 2HO_2 + CO$ | $j_{HCHO}$[b] | | |
| R4 | $CH_3CHO+h\nu\rightarrow CH_3O_2+HO_2+ CO$ | $j_{CH3CHO}$[b] | | |
| R5 | alkenes+$O_3\rightarrow$OH, $HO_2$, $RO_2$+products | | | |
| R5a | propene+$O_3\rightarrow$ products[c] | $1.0\times10^{-17}$ | ±20% | IUPAC |
| R5b | cis-but-2-ene+$O_3\rightarrow$ product[d] | $1.3\times10^{-16}$ | ±12% | IUPAC |
| R5c | 1-pentene+$O_3\rightarrow$ products[e] | $1.0\times10^{-17}$ | ±20% | MCMv3.3.1 |
| R5d | 2-hexene+$O_3\rightarrow$ products[f] | $1.1\times10^{-17}$ | ±20% | MCMv3.3.1 |
| R5e | isoprene+$O_3 \rightarrow$ products[g] | $1.3\times10^{-17}$ | ±10% | MCMv3.3.1 |
| R5f | α-pinene+$O_3 \rightarrow$ products[h] | $9.6\times10^{-17}$ | ±20% | IUPAC |
| **Radical interconversion reactions** | | | | |
| R6 | $HCHO+OH+O_2\rightarrow CO+H_2O+HO_2$ | $8.5\times10^{-12}$ | ±10% | IUPAC |
| R7 | $CO+OH+O_2\rightarrow CO_2+HO_2$ | $2.3\times10^{-13}$ | ±6% | IUPAC |
| R8 | $VOCs+OH+O_2 \rightarrow RO_2+H_2O$ | [j] | | |
| R9 | $RO_2+NO\rightarrow products+HO_2+NO_2$ | $8.6\times10^{-12}$ | ±30% | Jenkin et al. (2019) |
| R10 | $HO_2+NO\rightarrow OH+NO_2$ | $8.5\times10^{-12}$ | ±13% | IUPAC |
| R11 | $HO_2+O_3\rightarrow OH+2O_2$ | $2.0\times10^{-15}$ | ±29% | IUPAC |
| **Radical termination reactions** | | | | |
| R12 | $NO_2+OH\rightarrow HNO_3$ | $1.0\times10^{-11}$ | ±30% | IUPAC |
| R13 | $NO+OH\rightarrow HONO$ | $9.7\times10^{-12}$ | ±13% | IUPAC |
| R14 | $RO_2+NO\rightarrow RONO_2$ | $4.6\times10^{-13}$ | ±30% | Jenkin et al. (2019) |
| R15 | $RO_2+RO_2\rightarrow products$ | $3.5\times10^{-13}$ | ±50% | Jenkin et al. (2019) |
| R16 | $RO_2+HO_2\rightarrow ROOH+O_2$ | $2.3\times10^{-11}$ | ±50% | Jenkin et al. (2019) |
| R17 | $HO_2+HO_2\rightarrow H_2O_2+O_2$ | $4.5\times10^{-12}$[i] | ±20% | IUPAC |
| **Isoprene reactions** | | | | |
| R18 | isoprene + OH $\rightarrow$ products | $1.0\times10^{-10}$ | ±8% | IUPAC |
| R19 | isoprene–$RO_2$ (1,6-H shift) $\rightarrow$ products + OH | 0.01–0.06 $s^{-1}$ | | Peeters et al. (2014) |
| **Cl reactions** | | | | |
| R20 | $ClNO_2+h\nu\rightarrow Cl+NO_2$ | $j_{ClNO2}$[b] | | |
| R21 | $Cl_2+h\nu\rightarrow 2Cl$ | $j_{Cl2}$[b] | | |
| R22 | $VOCs+Cl\rightarrow RO_2+HCl$ | [j] | | |

used for the calculations.

[a] 1σ uncertainty
[b] Measured photolysis frequencies
[c] Yield for OH: 0.36, $HO_2$: 0.10, $RO_2$: 0.42 from Novelli et al. (2021)
[d] Yield for OH: 0.36, $HO_2$: 0.15, $RO_2$: 0.51 from Novelli et al. (2021)
[e] Yield for OH: 0.32, $HO_2$: 0.09, $RO_2$: 0.37 from Novelli et al. (2021)
[f] Yield for OH: 0.48, $HO_2$: 0.11, $RO_2$: 0.59 from Novelli et al. (2021)
[g] Yield for OH: 0.26, $HO_2$: 0.26 from Malkin et al. (2010)
[h] Yield for OH: 0.8 from Cox et al. (2020)
[i] at 1% water vapour mixing ratio
[j] Highly variable depending on the specific VOC.



## 2 Methodology

### 2.1 The JULIAC campaign

The Jülich Atmospheric Chemistry Project (JULIAC) campaign was conducted at Forschungszentrum Jülich (FZJ, 50.9° N, 6.4° E), Germany. The project consisted of four one-month long intensive campaigns studying atmospheric chemistry in ambient air in each season throughout 2019. The location is surrounded by a deciduous forest and is located in a rural environment near a town, Jülich (33,000 inhabitants), 25 km northeast, 40 km west, and 43 km southwest from three large cities, Aachen, Cologne and Düsseldorf, respectively. Therefore, ambient air is influenced by both biogenic and anthropogenic emission sources.

The investigation of the photochemistry was performed in the SAPHIR chamber, which was equipped with a large set of instruments measuring radicals, trace gases and aerosol (Table 2). The SAPHIR chamber has a cylindrical shape and is made of a double-wall Teflon (FEP) film. A slight overpressure (35 Pa) is

**Table 2**. Specification of instruments used in the JULIAC campaign for the analysis in this work.

| Species | Measurement technique | Time resolution | Limit of detection (1σ) | 1σ accuracy |
|---|---|---|---|---|
| OH | LIF | 270 s | $0.7 \times 10^6$ cm$^{-3}$ | 18% |
| OH | DOAS | 134 s | $0.8 \times 10^6$ cm$^{-3}$ | 6.5% |
| HO$_2$ | LIF | 47 s | $1 \times 10^7$ cm$^{-3}$ | 18% |
| RO$_2$ | LIF | 47s | $2 \times 10^7$ cm$^{-3}$ | 18% |
| OH reactivity ($k_{OH}$) | LP-LIF | 180 s | 0.2 s$^{-1}$ | 10% |
| Photolysis frequencies | Spectroradiometer | 60 s | | 18% |
| O$_3$ | UV photometry | 60 s | 0.5 ppbv | 2% |
| NO$_X$ (NO+NO$_2$) | Chemiluminescence | 60 s | NO: 20 pptv NO$_2$: 30 pptv | NO: 5 % NO$_2$: 7% |
| CO, CO$_2$, CH$_4$, H$_2$O | Cavity ring-down spectroscopy | 60 s | CO and CH$_4$: 1 ppbv CO$_2$: 25 ppbv H$_2$O: 0.1 % | 5% |
| HONO | LOPAP | 180 s | 5 pptv | 10% |
| HCHO | Cavity ring-down spectroscopy | 300 s | 0.1 ppbv | 10% |
| ClNO$_2$ | I-CIMS | 60 s | 2.8 pptv | 8.5% |
| VOCs | PTR-TOF-MS | 30 s | 15 pptv | 14% |
| | VOCUS PTR-TOF-MS | 30 s | | |
| Aerosol surface area | SMPS | 7 min | 10nm – 1µm | N/A |




maintained in the chamber and the space between the two films is permanently flushed with pure nitrogen
(Linde, purity: > 99:99990 %) to prevent outside air penetrating the inner chamber. The chamber is
equipped with a shutter system allowing the air to be either shielded from or exposed to solar radiation.
In the JULIAC campaign, ambient air was sampled at a high flow rate of 660 m$^3$ h$^{-1}$ from 50 m high inlet
line (104 mm inner diameter, SilcoNert® coated stainless steel) by means of an oil-free turbo blower
(Aerzener Maschinenfabrik, AERZEN Turbo G3 Typ: TB 50-0.6 S). Large particles (>10 μm diameter)
were removed by a SilcoNert® coated cyclone (LTG, ZSB-6). The temperatures in the inlet line and cyclone
were controlled to be slightly higher than ambient temperature (+1 to 2 ˚C) to avoid water vapor
condensation in the inlet system. A 3/2-way valve directed part of the air (flow rate of 250 m$^3$ h$^{-1}$) into the
chamber. Two fans inside the chamber ensured fast mixing on a time scale of a few minutes. As a result,
the chamber behaved as a continuously stirred photochemical flow reactor with a mean residence time of
air of 1.1 h. During the transition time of 3.5 s from the tip of the inlet to the SAPHIR chamber, atmospheric
RO$_X$ radicals are lost on walls, but concentrations are rapidly re-established in the sampled ambient air
inside the sunlit chamber.
The use of the chamber as a flow reactor has advantages compared to field measurements in the open air.
Perturbations of the studied chemistry due to local emissions of VOCs or NO$_X$ can be avoided. Transient
fluctuations of reactants in the sampled air, for example due to spikes of NO from passing cars, are
smoothed out in the chamber. Due to the homogeneous mixing, instruments connected to the chamber
measure the same air composition and segregation effects on reaction rates are insignificant.
The air composition could be influenced by the inlet line and chamber surfaces. As the whole inlet line is
heated and chemically inert due to the SilcoNert® coating, no relevant wall loss or desorption of trace gases
is expected from the inlet. This assumption was confirmed by comparing OH reactivity measured at several
positions of the inlet line. No significant differences were found between measurements, if the air was either
sampled upstream of the cyclone or downstream of the blower. Wall losses of trace gases (VOCs, NOx, O$_3$)
inside the SAPHIR chamber were found to be negligible in previous experiments (e.g., Kaminski et al.,
2017, Rolletter et al., 2020).
Nitrous acid (HONO) and formaldehyde (HCHO) are known to be emitted from the chamber film when it
is exposed to solar radiation (Rohrer et al. (2005)). These emissions significantly increase the
concentrations of HONO and HCHO in the chamber. Due to the transmission through the Teflon film and
shading from construction elements of the chamber, the absolute actinic flux density is reduced by 20 to
40 % compared to outside the chamber. It is worth noting, however, that the relative spectral distribution
of the solar radiation is not changed by the transmission through the chamber film (Bohn and Zilken, 2005).
The floor underneath the chamber is heated by the solar radiation. Although it is not in direct contact to the
foil, the air temperature in the chamber was on average 0.7°C higher during winter and autumn and 1.9°C
higher during spring and summer than the temperature outside of the chamber at daytime. Since
photochemistry was studied in the chamber, all data of chemical and physical conditions shown in this work
refer to conditions inside the chamber.
The measurements in the campaign were at least once a week interrupted for calibration and maintenance
of instruments. Some days were also excluded from the analysis in this work because the chamber shutter
system was kept closed to protect the chamber film during bad weather from strong wind gusts and/or



precipitation. Reference experiments with clean synthetic air were performed to investigate possible
changes in the strength of chamber emissions and to check for instrumental backgrounds.

**2.2 Instrumentation**
**2.2.1 OH, HO$_2$ and RO$_2$ radical and OH reactivity ($k_{OH}$) measurements**
OH, HO$_2$, and RO$_2$ radicals were measured by the FZJ – LIF which included a newly developed chemical
modulation reactor (CMR) for interference-corrected measurements of OH radicals (Cho et al., 2021). The
signals of the instrument were calibrated against well-defined radical concentrations that were produced
from water photolysis in synthetic air at a wavelength of 185nm using radiation of a mercury lamp. A
detailed description of the LIF instrument and its calibration can be found in previous publications (Holland
et al., 2003; Fuchs et al., 2008; Fuchs et al., 2011; Fuchs et al., 2012).
Shortly, the OH radical is sampled through a nozzle with a 0.4 mm diameter pinhole and is excited by a
pulsed laser at a wavelength of 308 nm in a low-pressure (4 hPa) fluorescence cell. The emitted resonant
fluorescence is detected with a time delay by a time-gated micro-channel plate detector (MCP). In the
JULIAC campaign, a chemical modulation reactor (CMR) was implemented on top of the OH cell to
quantify potential interferences. This is achieved by periodically removing ambient OH by an OH scavenger
that is injected in the reactor (propane, Air Liquide, purity>99.95%, (5.0±0.1) % mixture in nitrogen) before
the air enters the fluorescence cell. During the campaign, the observed interference could be fully explained
by the well-characterized interference from the photolysis of ozone in humid air inside the detection cell.
No evidence for an unexplained interference was found (Cho et al., 2021). The limit of detection for OH
was $0.7 \times 10^6$ cm$^{-3}$ and the accuracy was 18 % (1σ).
OH radical concentrations were also measured by differential optical absorption spectroscopy (DOAS)
using a multiple folded light path for absorption inside along the chamber. The DOAS technique is a
calibration-free technique (Hausmann et al., 1997; Schlosser et al., 2007; Schlosser et al., 2009). The limit
of detection was $0.8 \times 10^6$ cm$^{-3}$ and the 1σ–accuracy was 6.5 %. Due to a technical laser problem, the
DOAS instrument was not available in spring.
HO$_2$ radicals were detected by the LIF instrument in a separate detection cell, where HO$_2$ is chemically
converted to OH radicals in the reaction with NO (Air Liquide, 1% NO in N$_2$, purity > 99.5 %) that is
injected in the fluorescence cell (Fuchs et al., 2011). During the JULIAC campaign, two different
concentrations ($2.5 \times 10^{13}$ cm$^{-3}$ and $1.0 \times 10^{14}$ cm$^{-3}$) of NO in the fluorescence cell were used to
observe possible interference from specific RO$_2$ radicals as highlighted by Fuchs et al. (2011). No difference
between HO$_2$ measurements at high and low NO concentrations was found suggesting that there was no
significant interference from RO$_2$.
In addition, the sum of OH, HO$_2$, and RO$_2$ (RO$_X$) was measured by the RO$_X$-LIF system. Air is sampled
into a chemical converter (pressure of ~ 25 hPa), where a mixture of NO (Air Liquide, 500 ppmv NO in N$_2$,
purity > 99.5%) and CO (Air Liquide, 10% CO in N$_2$, purity > 99.997%) is injected. The NO converts RO$_2$
radicals to HO$_2$ radicals and CO converts OH radicals formed from the reaction of HO$_2$ radicals with NO
back to HO$_2$. Therefore, an equilibrium between OH and HO$_2$ is established. Concentrations are chosen, so



that the equilibrium is on the side of $HO_2$. In a low-pressure cell downstream of the converter $HO_2$ radicals
are converted to OH radicals by injecting excess NO (Air Liquide, pure NO, purity>99.5%) (Fuchs et al.,
2008) that shifts the equilibrium between OH and $HO_2$ to OH. The $RO_2$ concentration is obtained from the
difference between the sum measurement of $RO_X$ and measurements of OH and $HO_2$ concentrations in the
other two detection cells. The $RO_2$ detection sensitivity was calibrated for methyl peroxy radicals ($CH_3O_2$)
which are produced from the reaction of OH with methane ($CH_4$) in the calibration system. The resulting
calibration is also applicable to the majority of other atmospheric alkyl peroxy radicals (Fuchs et al., 2008;
Fuchs et al., 2011).
The signals in the $HO_2$ and $RO_2$ detection systems contain a background signal observed when NO is
injected into the detection cells, even if no radicals are present in the air sampled. The background signal
can be characterized when the inlet of the detection system is overflown with synthetic air, which is part of
the calibration procedures. During JULIAC the background varied from calibration to calibration and was
often larger than the smallest signals measured in ambient air from the chamber (Table S1). The highest
background signals obtained from calibrations is therefore regarded as an upper limit and the variability is
considered as an additional uncertainty in the measured $HO_2$ and $RO_2$ concentrations. $HO_2$ and $RO_2$
background signals, which are subtracted in the evaluation of $HO_2$ and $RO_2$ measurements, were taken from
reference experiments in the dark clean chamber, when no $HO_2$ or $RO_2$ radicals are expected. The subtracted
signals for each period are available in Table S1and in most cases were equivalent to concentrations lower
than $1 \times 10^7$ cm$^{-3}$ for both $HO_2$ and $RO_2$ measurements.
The total OH reactivity ($k_{OH}$), the inverse of the chemical lifetime of OH radicals, was measured in ambient
air by a laser-flash photolysis LIF instrument (Lou et al., 2010; Fuchs et al., 2017). A high concentration of
OH radicals is produced by flash photolysis (266 nm, 1 Hz repetition rate) of ozone in humid air (Reaction
R2) in a flow tube that is on top of an OH fluorescence cell. The pseudo first-order decay of OH in the
chemical reactions with atmospheric reactants is measured, giving directly the OH reactivity.

### 2.2.2 Other trace gases, aerosol properties and photolysis frequencies measurements

A comprehensive set of instruments operated during the JULIAC campaign (Table 2) analyzed the air
composition inside the chamber. Photolysis frequencies inside the chamber were derived from the solar
actinic flux densities measured by a spectroradiometer mounted on the roof of the nearby institute building
(Bohn et al., 2005; Bohn and Zilken, 2005). Formaldehyde (HCHO) was detected by cavity ring-down
spectroscopy (Picarro, G2307, Glowania et al. (2021)). NO and $NO_2$ were measured by chemiluminescence
(Eco Physics, TR780). In addition, HONO was measured by long-path absorption photometry (LOPAP,
Kleffmann et al. (2006); Häseler et al. (2009)), CO, $CO_2$, $CH_4$, and $H_2O$ by cavity ring-down spectroscopy
(Picarro, G2401), and $O_3$ by UV absorption (Ansyco-41M and Thermo scientific-49I). Volatile organic
compounds (VOCs) were detected by a proton-transfer-reaction time-of-flight mass spectrometer (PTR-
TOF-MS, Ionicon) (Jordan et al., 2009) and a VOCUS PTR-TOF-MS instrument (Aerodyne). The VOCs
included in this study are listed in Table S2 and include isoprene and some carbonyl compounds. Total
aerosol surface area was determined from measurements by a scanning mobility particle sizer (SMPS). In
the summer and autumn periods, nitryl chloride ($ClNO_2$) was detected by a chemical ionization mass
spectrometer using iodine as reagent ion (I-CIMS) (Sommariva et al., 2018; Tan et al., 2022).
In addition to measurements in the chamber, concentrations of $O_3$ and $NO_X$ were also measured in the inlet
system before the air flowed into the SAPHIR chamber. For these measurements, a combined system (Eco



Physics, CraNO$_X$) consisting of an ozone photometer and a chemiluminescence instrument for NO$_X$ was
deployed. Measurements were used to determine the photochemical ozone production in the JULIAC
campaign. Further description of the measurement set-up and concept of the evaluation will be discussed
in details in a further publication.

**2.3 Chemical budget calculations**
A chemical budget analysis, similar as in Tan et al. (2019) and Whalley et al. (2021), was applied for OH,
HO$_2$, RO$_2$ and the sum of all three radicals (RO$_X$) to the data set from the JULIAC campaign. All reactions
typically considered to be relevant for the generation and destruction of these radicals are considered (Table
1). Rate constants and their uncertainties were mainly taken from IUPAC recommendations (Atkinson et
al., 2004; Atkinson et al., 2006; Cox et al., 2020) or more recent studies. If not otherwise specified, radical
production and destruction rates are calculated from measured concentrations of reactants.
**2.3.1 Chemical budget of OH radicals**
The production rate of OH radicals includes primary production reactions (Reaction R1, R2 and R5) and
radical interconversion reactions (Reaction R10 and R11):
$P_{OH} = j_{HONO}[HONO] + \varphi_{OH}j_{O^1D}[O_3] + k_{10}[NO][HO_2] + k_{11}[O_3][HO_2]$
$+\Sigma\{\varphi_{OH}{}^i k_5^i[alkene]^i[O_3]\} + P_{OH,Isop.}$     (1)
Here, $\varphi_{OH}$ is the effective OH yield of the ozone photolysis including the reaction of excited oxygen atoms
O($^1$D) with H$_2$O producing two OH radicals. $\varphi_{OH}{}^i$ is the OH yield of the ozonolysis reaction of alkenes,
and $k_5^i$ represents the rate constants of the corresponding reactions.
$P_{OH,Isop}$ is the effective production of OH radicals from unimolecular reactions (1,6-hydrogen shift reactions)
of isoprene-RO$_2$ radicals (Z-δ-RO$_2$-I and II, Peeters et al. (2014)) and the subsequent chemistry of products..
As there was no measurement of speciated RO$_2$ radicals, isoprene-RO$_2$ radical concentrations are estimated
from steady-state conditions considering their production from the reaction of isoprene with OH and their
destruction in bimolecular reaction (reaction rate $k_{bi}$) and unimolecular reactions (bulk reaction rate $k_{bulk\,1,6-}$
$_H$ as defined in Peeters et al. (2014)):
$[RO_2(isop.)]_{SS} = \dfrac{k_{18}[Isoprene][OH]}{k_{bi}+k_{bulk\,1,6-H}}$     (2)
$k_{bi} = (k_9 + k_{14})[NO] + k_{15}[RO_2] + k_{16}[HO_2]$     (2a)
Bimolecular loss reactions include reactions with NO (Reaction R9 and R14), RO$_2$ (Reaction R15) and HO$_2$
(Reaction R16). The OH production from isoprene-RO$_2$ isomerization reactions is simplified in the
calculation of the total OH production in this work by assuming that each isomerization reaction produces
rapidly one OH radical from the subsequent reactions of products such as photolysis of hydroxy-peroxy
aldehyde (HPALD). In this case, the radical production rate is equal to the loss rate of the isoprene-RO$_2$
due to isomerization reactions ($D_{Z-\delta-R_2,Isop.}$):





$P_{OH,Isop.} = D_{Z-\delta-RO_2,\ Isop.} = k_{bulk\ 1,6-H}\ [RO_2(isop.)]_{SS}$                                   (4)
The total loss rate of OH radicals for the chemical budget analysis is determined by the product of the total
OH reactivity ($k_{OH}$) and the OH radical concentration:
$D_{OH} = k_{OH}[OH]$                                                                                              (5)

### 2.3.2 Chemical budget of HO₂ radicals

The production rate of $HO_2$ radicals includes primary reactions (Reaction R3, R4 and R5) and
interconversion reactions (Reaction R6, R7 and R9, Table 1):
$$P_{HO_2} = 2\,j_{HCHO}[HCHO] + k_6[HCHO][OH] + k_7[CO][OH] + k_9[NO][RO_2]$$
$$+\Sigma\{\varphi_{HO_2}{}^i k_5^i[alkene]^i[O_3]\}$$                                                            (6)
Here, the photolysis frequency of HCHO ($j_{HCHO}$) include only paths generating radicals. $\varphi_{HO_2}{}^i$ is the $HO_2$
yield from the ozonolysis of alkenes. The reactions of OH with $H_2$ and $O_3$ are not considered due to their
negligible contributions to the $HO_2$ production.
The loss rate of $HO_2$ is determined by the reactions with NO (Reaction R10), $O_3$ (Reaction R11), $RO_2$
(Reaction R16) and $HO_2$ (Reaction R17):
$D_{HO_2} = (k_{10}[NO] + k_{11}[O_3] + k_{16}[RO_2] + 2k_{17}[HO_2])[HO_2]$                                       (7)
The reaction of $HO_2$ radicals with $NO_2$ is not included as the thermal decomposition of peroxynitric acid
($HO_2NO_2$) forming back $HO_2$ radicals and $NO_2$ is instantaneous for the temperatures experienced during
the JULIAC campaign.
In a sensitivity calculation (Section 4.2.3), potential loss of $HO_2$ due to heterogeneous uptake on aerosol is
investigated. The first order loss rate ($k_{het.}$) can be described as:
$k_{het.} = \dfrac{\gamma_{eff.} \cdot \nu_{HO_2} \cdot [AS]}{4}$                                                  (8)
$\nu_{HO_2}$ is the mean molecular velocity of $HO_2$ (4.44 ×10⁵ cm s⁻¹ at 298 K), [AS] is the measured aerosol
surface area concentration, and $\gamma_{eff.}$ is the effective uptake coefficient.

### 2.3.3 Chemical budget of RO₂ radicals

Primary sources of $RO_2$ radicals include all oxidation reactions of VOCs with OH, Cl, $NO_3$ radicals and $O_3$.
As the number of measured VOC species in this study was limited (Table S2) and because it is generally
difficult to capture the entire spectrum of atmospheric VOCs (Goldstein and Galbally, 2007; Lou et al.,
2010), the measured total OH reactivity ($k_{OH}$) can be used to calculate the $RO_2$ radicals production from the
reactions of VOCs with OH. First, the contributions from CO, NO, $NO_2$, HCHO and $O_3$ is removed from
the measured OH reactivity as these species do not form $RO_2$ radicals in the reaction with OH. It is then
assumed that the remaining fraction can be attributed to organic compounds (VOC reactivity ($k_{VOC}$))
including measured and unmeasured VOCs, which produce $RO_2$ radicals in their reaction with OH




In addition, $RO_2$ production from ozonolysis needs to be included. In this work, only the reactions of
measured organic compounds are considered. The contribution to the $RO_2$ production from the oxidation
of VOCs by the $NO_3$ radical was negligible during daytime due to the low VOC load (low OH reactivity),
so that $NO_3$ destruction by photolysis and reaction with NO dominated.
Reactions of chloride (Cl) also produce $RO_2$ radicals, but the concentration was not measured in the
JULIAC campaign. However, one of the most important precursor species, nitryl chloride ($ClNO_2$), was
detected during the campaign (except in spring, Tan et al. (2022)). $ClNO_2$ can accumulate during nighttime,
but it is photolyzed after sunrise yielding $NO_2$ and Cl atoms (Reaction R20). Assuming as an upper limit
that each Cl atom reacts with a VOCs (Tanaka et al., 2003), the $RO_2$ production rate from Cl radicals can
be calculated as:
$$P_{RO_2,Cl} = j_{ClNO_2}[ClNO_2] \tag{9}$$
The total $RO_2$ production rate is then calculated as:
$$P_{RO_2} = k_{VOC}[OH] + \Sigma\left(\varphi_{RO_2}{}^i k_{R5}^i [alkene]^i [O_3]\right) + P_{RO_2,Cl} \tag{10}$$
Here, $\varphi_{RO_2}{}^i$ is the $RO_2$ yield from the ozonolysis of alkenes species (Table 1).
With respect to the destruction rate of $RO_2$, its reactions with NO, $HO_2$, and other $RO_2$ and unimolecular
reactions of specific isoprene-$RO_2$ radicals ($D_{Z-\delta-R\ _2,Isop.}$) (Eq. 4) are considered in this work:
$$D_{RO_2} = ((k_9+k_{14})[NO] + 2k_{15}[RO_2] + k_{16}[HO_2])[RO_2] + D_{Z-\delta-R\ _2,\ Isop.} \tag{11}$$
**2.3.4 Chemical budget of $RO_X$ radicals**
In the chemical budget of the sum of OH, $HO_2$ and $RO_2$ ($RO_X$), inter-radical conversion reactions cancel
out and only initiation and termination reactions are included. Therefore, the $RO_X$ radical budget analysis
allows to investigate if primary radical source reactions or termination processes are missing in the chemical
mechanism used (Table 1).
The production rate of the $RO_X$ radicals is given by the sum of rates from radical initiation reactions
(Reaction R1-R5, R20-R22, Table 1):
$$P_{RO_x} = j_{HONO}[HONO] + \varphi_{OH}j_{O^1D}[O_3] + 2j_{HCHO}[HCHO]$$
$$+ \Sigma\left((\varphi_{OH}^i + \varphi_{HO_2}^i + \varphi_{RO_2}^i)k_5^i[alkene]^i[O_3]\right) + P_{RO_2,Cl} \tag{12}$$
Radicals can be additionally produced from the photolysis of other oxygenated organic compounds
(OVOCs, e.g., Reaction R4) not included in Eq. 12. Their potential impact is further discussed in Section
335 4.2.2.

The loss rate of the $RO_X$ radical is calculated by the sum of rates from radical termination reactions
(Reaction R12-R17):
$$D_{RO_x} = (k_{13}[NO] + k_{12}[NO_2])[OH] + k_{14}[NO][RO_2] + 2k_{15}[RO_2]^2 + 2k_{16}[HO_2][RO_2] + 2k_{17}[HO_2]^2$$
$$\tag{13}$$



**2.3.5 Uncertainties in the calculated production and destruction rates**
The uncertainty of each production or loss rate is calculated by Gaussian summation of the $1\sigma$ errors of the
measured quantities (Table 2) and the uncertainties of the reaction rate constants (Table 1).
For reactions of $RO_2$ with NO (Reaction R9, R14), $HO_2$ (Reaction R16) and $RO_2$ (Reaction R15), generic
rate constants are used for the sum of $RO_2$ radicals (Table 1, Jenkin et al. (2019)). Rate constants of the NO
reaction with $RO_2$ derived from hydrocarbons ($<C_5$) and with oxygenated peroxy radicals range from
$7.7 \times 10^{-12}$ cm$^3$ s$^{-1}$ to $1.1 \times 10^{-11}$ cm$^3$ s$^{-1}$ (Jenkin et al., 2019). The $1\sigma$-uncertainty of the rate constants
varies from 6 to 30 %. In the error calculations here, an upper limit value of 30 % is applied. However, for
reactions of $RO_2$ with $HO_2$ and with $RO_2$, the range of rate constants varies by more than an order of
magnitude. In the calculations, an uncertainty of 50% is used for the reaction rate constants of $RO_2$ with
$HO_2$ and with $RO_2$.
As there are no measurements of speciated $RO_2$ radicals, a yield of 5% for the formation of organic nitrates
is assumed for all $RO_2$ but the yield can vary between 1% for methyl peroxy radicals ($CH_3O_2$) and more
than 20 % for $RO_2$ from monoterpene species. This simplification can introduce systematic errors in the
calculations (Section 4.2.1).

**2.4 Odd oxygen production rate**
In the troposphere, ozone is formed exclusively by the oxidation of NO to $NO_2$ through reaction with $RO_2$
(Reaction R9) and $HO_2$ (Reaction R10), followed by $NO_2$ photolysis (Fishman and Carney, 1984; Sillman
et al., 1990; Kleinman et al., 2002).
During the day, the photolysis of $NO_2$ and the back reaction of NO with $O_3$ form a rapid photochemical
equilibrium between $O_3$ and $NO_2$. The sum of $O_3$ and $NO_2$ is therefore defined as odd oxygen ($O_X$) (Han et
al., 2011; Goldberg et al., 2015). The relative composition of $O_X$ depends on the $NO_2$ photolysis frequency
and the NO concentration. For the conditions of the spring and summer periods in the JULIAC campaign,
$O_X$ consisted predominantly (> 85%) of $O_3$.
In this work, the net production rate of $O_X$ ($P_{Ox}$) was determined experimentally from the increase of $O_X$ in
the sunlit SAPHIR chamber. Furthermore, measurements of radicals and $NO_X$ were used to calculate $P_{Ox}$
from the rate of $O_X$ formation reactions (Reaction R9, R10), and $O_X$ loss by the reaction of $NO_2$ with OH
(Reaction R12) (Mihelcic et al., 2003; Cazorla et al., 2012; Niether et al., 2022)):
$$P_{Ox,net} = k_9[NO][RO_2] + k_{10}[NO][HO_2] - k_{12}[NO_2][OH] \qquad (14)$$
This calculation neglects minor $O_x$ destruction processes such as the reaction of $O_3$ with $NO_2$, OH, $HO_2$ or
Cl since they did not play a notable role during the day in this campaign.
**3 Results**
**3.1 Data quality of radical measurements**





Performing measurements in the SAPHIR chamber allowed to test the accuracy of radical measurements
in different ways that are typically not available in field experiments. First, OH radicals was measured by
2 independent instruments, the OH-DOAS and LIF instruments (Cho et al., 2021). Second, the $O_X$
production rate calculated from measured concentrations of $HO_2$ and $RO_2$ could be compared to the
observed increase of $O_X$ concentrations in the chamber, which can be solely attributed to chemical reactions.
This is possible, because other factors typically impacting the $O_X$ concentration in field experiments such
as transportation processes are not effective.
OH concentrations were measured by the LIF instrument applying the chemical modulation scheme and
the DOAS in the winter, summer and autumn periods of the campaign. As OH concentrations were close
to the limit of detection in autumn and winter, a meaningful comparison of measurements was only possible
for the summer period. A detailed comparison of measurements can be found in Cho et al. (2021). In general,
the OH measurements of the two instruments agreed within their measurement errors (Table 1) giving a
slope of 1.1±0.02 in a linear regression analysis. The good agreement confirms that the newly developed
chemical modulation system of the LIF instrument allowed for interference-free OH concentration
measurements for conditions of the campaign. Only in the period from 22 to 26 August, which was
characterized by exceptionally high temperatures (30 to 40°C), OH concentrations measured by the LIF
instrument were systematically higher by 25% than those measured by the DOAS instrument for unknown
reasons (Cho et al., 2021). OH concentrations measured by the DOAS instrument were used for the analysis
of the radical budgets in this period.
Net $O_X$ production rates were determined from the measured increase of $O_X$ concentrations in the chamber
and compared to calculations from the turnover rates of $HO_2$ and $RO_2$ reactions with NO. This calculation
takes also the $NO_2$ loss due to its reaction with OH into account (Eq. 14). The odd oxygen production rate
did not exceed 1 ppbv h$^{-1}$ in winter and autumn due to the general low photochemical activity in these
seasons. In spring and summer, the $O_X$ production rate showed clear diurnal variations with noontime
maxima that reached up to 16 ppbv h$^{-1}$. In these seasons, both methods for determining the $O_X$ production
rate agreed within ±15 % (1σ). Observed discrepancies were less than 1 ppbv h$^{-1}$, when NO mixing ratios
were lower than 1 ppbv, but reached values of 3 ppbv h$^{-1}$ for NO mixing ratios of  3 - 4 ppbv NO. The
largest discrepancy of 8.5 ppbv h$^{-1}$ was found in the morning on 29 April, when the NO mixing ratio
exceeded 9 ppbv. High NO values suppressed $HO_2$ and $RO_2$ concentrations to values below $2.0 \times 10^7$ cm$^{-3}$,
which is within the range of the background corrections for the $HO_2$ and $RO_2$ measurements (Table S1).
Under these conditions, an erroneous background subtraction may have caused the observed discrepancies.










**3.2 Meteorological and chemical conditions during the JULIAC campaign**

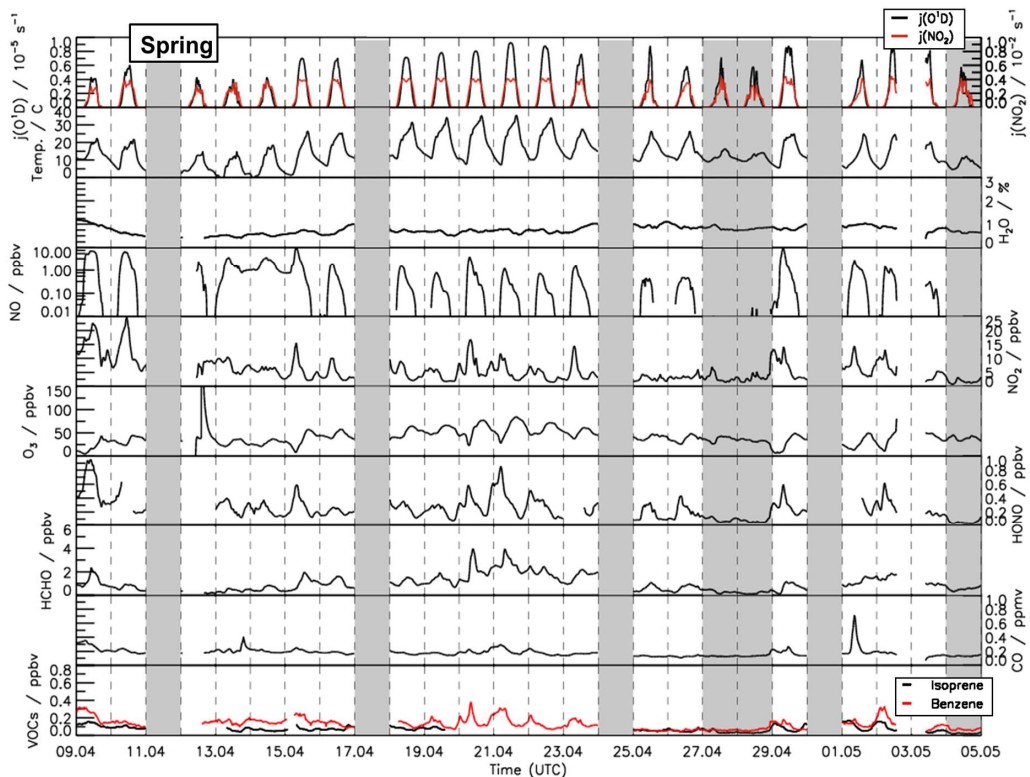

**Figure 1**: Time series of temperature and trace gas concentrations during the spring period of the JULIAC campaign (Cho et al., 2022). Vertical dashed lines denote midnight. Grey shaded areas indicate calibration days, when no measurements were done and days when the chamber roof was closed due to bad weather conditions.

A broad range of meteorological and chemical conditions was encountered during the JULIAC campaign.
During the winter and autumn periods (Fig. S1 and S2), the sky was often overcast and it rained frequently.
Temperatures were generally below 10°C and the photolysis frequencies of ozone ($j_{O1D}$) and nitrogen
dioxide ($j_{NO2}$) mostly remained below $1.5 \times 10^{-6}$ s$^{-1}$ and $2 \times 10^{-3}$ s$^{-1}$, respectively. During spring and
summer, temperatures in the chamber were up to 35°C in mid-April and 40°C between 24 and 31 August
(Fig. 1 and 2). Photolysis frequencies in the chamber were $1 \times 10^{-5}$ s$^{-1}$ ($j_{O1D}$) and $4 \times 10^{-3}$ s$^{-1}$ ($j_{NO2}$).
The air was sampled at all times from 50 m above ground. The temperature at different heights measured
between 5 m and 120 m at a meteorological tower near the SAPHIR chamber showed that the air was well
mixed within this height range during the day. Therefore, it can be assumed for the chemical composition
of the air sampled into the chamber to be representative for the air within the atmospheric boundary layer.
At night, vertical temperature profiles showed atmospheric stratification below 100 m. The air at 50 m can





be assumed to be isolated from the ground and therefore not being affected by surface emissions or
deposition on surfaces at the ground.
Overall, relatively clean air was sampled during the whole JULIAC campaign indicated by CO and NO
mixing ratios below 0.3 ppmv and 2 ppbv, respectively. Concentrations of anthropogenic organic
compounds (e.g. benzene and toluene) were low with mixing ratios of less than 0.5 ppbv. Even though the
measurement site is surrounded by a deciduous forest, the concentrations of biogenic organic compounds
such as isoprene and monoterpenes were also low (median 0.8 ppbv and 0.15 ppbv, respectively) compared
to previously reported values measured on the campus of FZJ in summer, when isoprene concentrations
ranged between 0.5 to 4 ppbv (Komenda et al., 2003; Spirig et al., 2005; Kanaya et al., 2012). A possible
reason for the low values could be damages of trees from severe droughts in the previous year (BMEL,
433    2021).


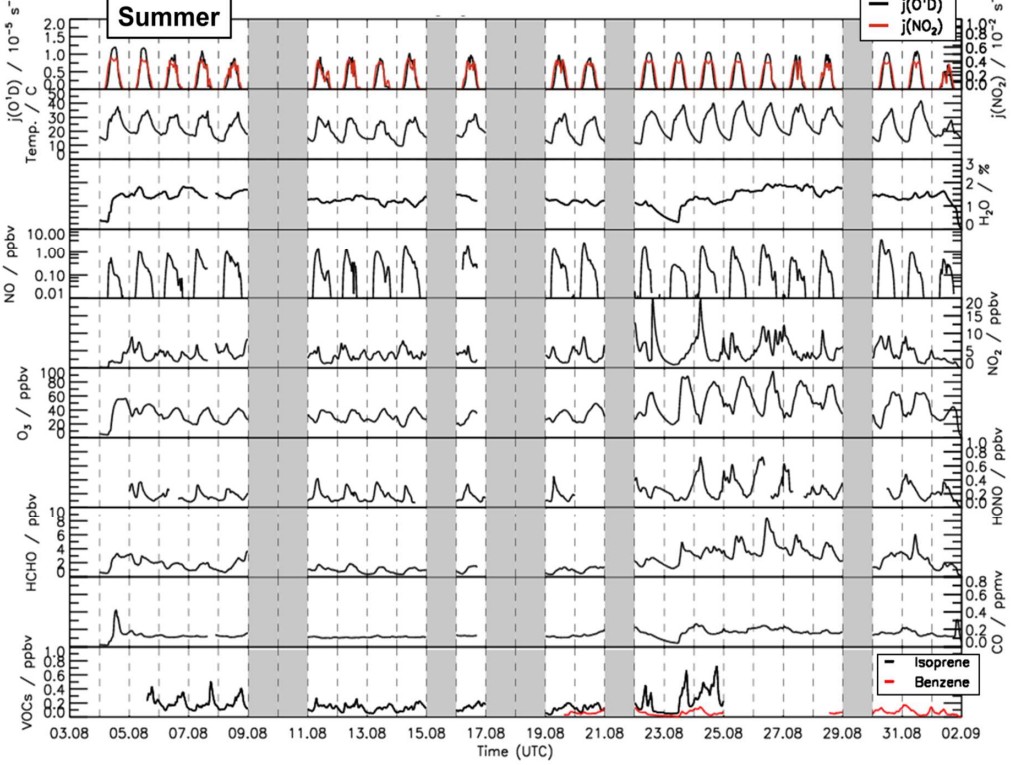

**Figure 2**: Time series of temperature and trace gas concentrations during the summer period of the
JULIAC campaign (Cho et al., 2022). Vertical dashed lines denote midnight. Grey shaded areas
indicate calibration days, when no measurements were done and days when the chamber roof was
closed due to bad weather conditions.




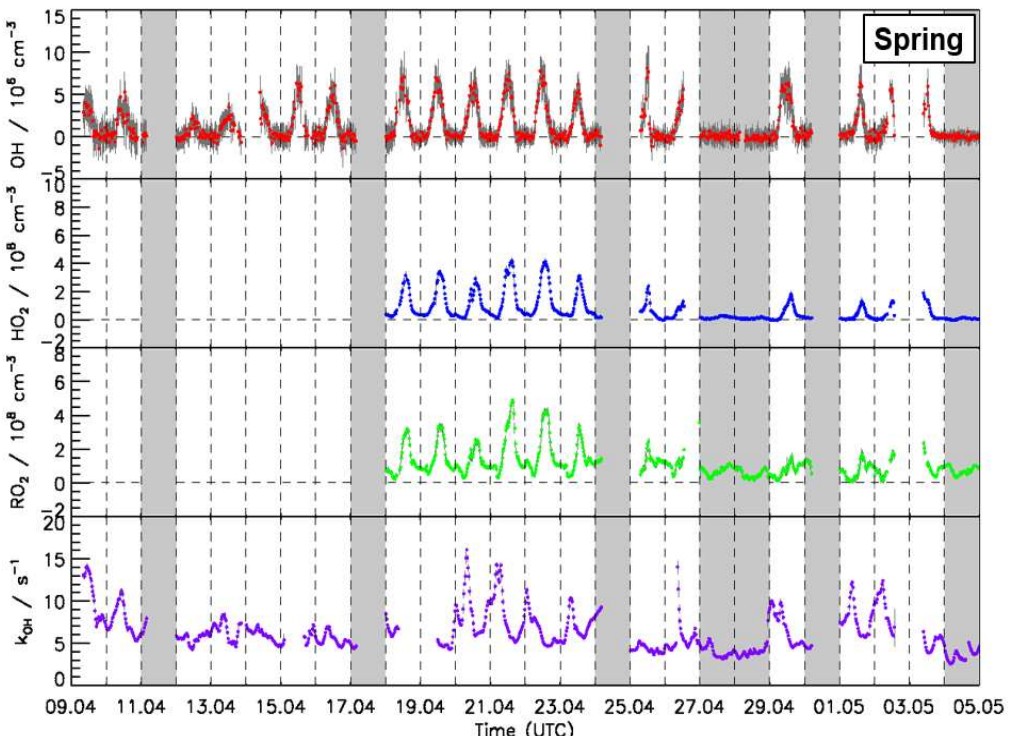

**Figure 3:** Time series of OH, HO$_2$, and RO$_2$ radical concentration measured by the FZJ-LIF-CMR instrument and measurements of the OH reactivity ($k_{OH}$) measured in the spring period of the JULIAC campaign (Cho et al., 2022). Vertical dashed lines denote midnight. Grey shaded areas indicate calibration days when no measurements were done and days when the chamber roof was closed due to bad weather conditions.

**3.3 OH, HO$_2$, and RO$_2$ radical concentrations and OH reactivity during winter and autumn periods of the JULIAC campaign**

During winter (Fig. S3) and autumn (Fig. S4), daytime OH radical concentrations were below $1 \times 10^6$ cm$^{-3}$, mainly due to a low primary radical production. Daytime peroxy radical (HO$_2$ and RO$_2$) concentrations during these periods were also very low with average values below $2 \times 10^7$ cm$^{-3}$ (Fig. S5) close to the limit of detection of RO$_2$ radicals (Table 2) and within the uncertainty of the background corrections for HO$_2$ and RO$_2$ (Table S1). During winter and autumn, HO$_2$ concentrations typically increased in the morning and reached peak concentrations of $2 \times 10^7$ cm$^{-3}$ at noon. Concentrations decreased in the evening and night with minimum values right before sunrise. In contrast, nighttime RO$_2$ concentrations increased to values between 3 to $4 \times 10^7$ cm$^{-3}$ after sunset, when the chemical loss due to their reaction with NO became negligible, while RO$_2$ radicals were still produced from reactions of VOC with NO$_3$ and O$_3$. NO concentrations were essentially zero at that time, because NO production by the photolysis of NO$_2$ stopped and NO rapidly reacted with ozone. RO$_2$ radical concentrations decreased in the morning to values that



were similar to that of HO₂ radicals as can be expected for conditions with high NO mixing ratios, which
lead to a fast loss of RO₂ and HO₂ in their reactions with NO.

The measured OH reactivity ($k_{OH}$) ranged between 4 and 33 s$^{-1}$ during winter and autumn periods. The
highest value was observed on 21 January, when a highly polluted plume containing 50 ppbv of NO was
sampled.

The measured OH reactivity can be compared to OH reactivity calculated by summing up the product
between measured OH reactant concentrations and their reaction rate constants with the OH radical. On
average, 1.3 s$^{-1}$ (18 %) of the measured OH reactivity could not be explained by the measured OH reactants
during the winter and autumn periods (Fig. S5). NO$_X$, CH₄, CO, and VOCs contributed approximately 43,
3, 20 and 13 %, respectively, to the measured OH reactivity.

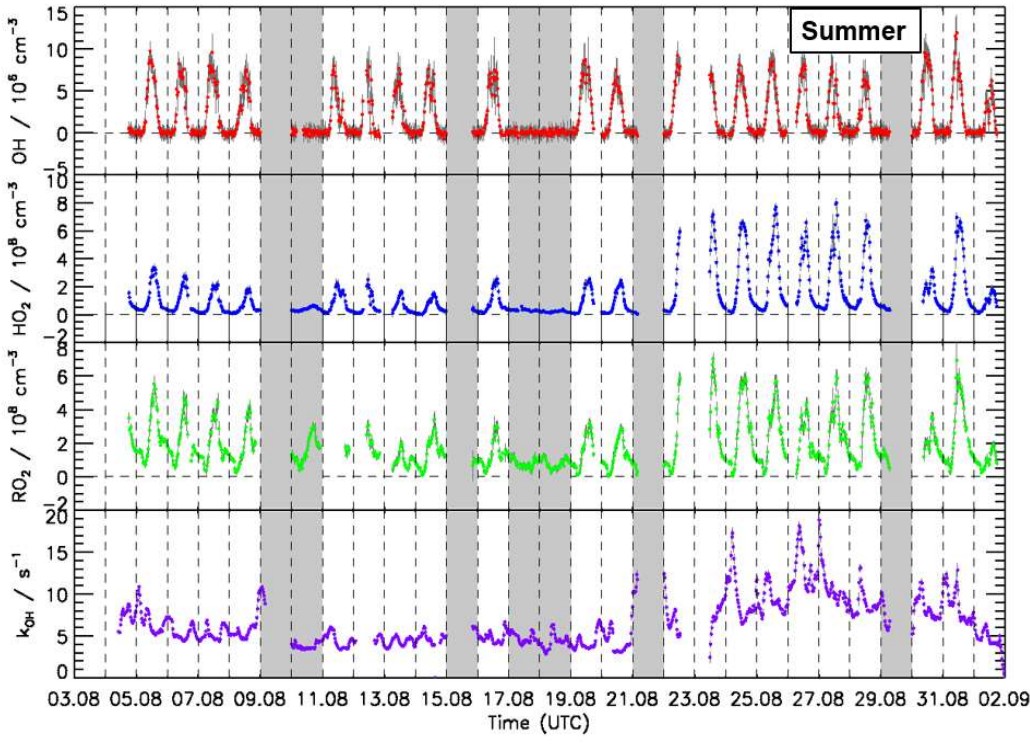

**Figure 4**: Time series of OH, HO₂, and RO₂ concentration measured by the FZJ-LIF-CMR instrument and
measurements of the OH reactivity ($k_{OH}$) in the summer period of the JULIAC campaign (Cho et al., 2022).
Vertical dashed lines denote midnight. Grey shaded areas indicate calibration days when no measurements
were done and days when the chamber roof was closed due to bad weather conditions.

458



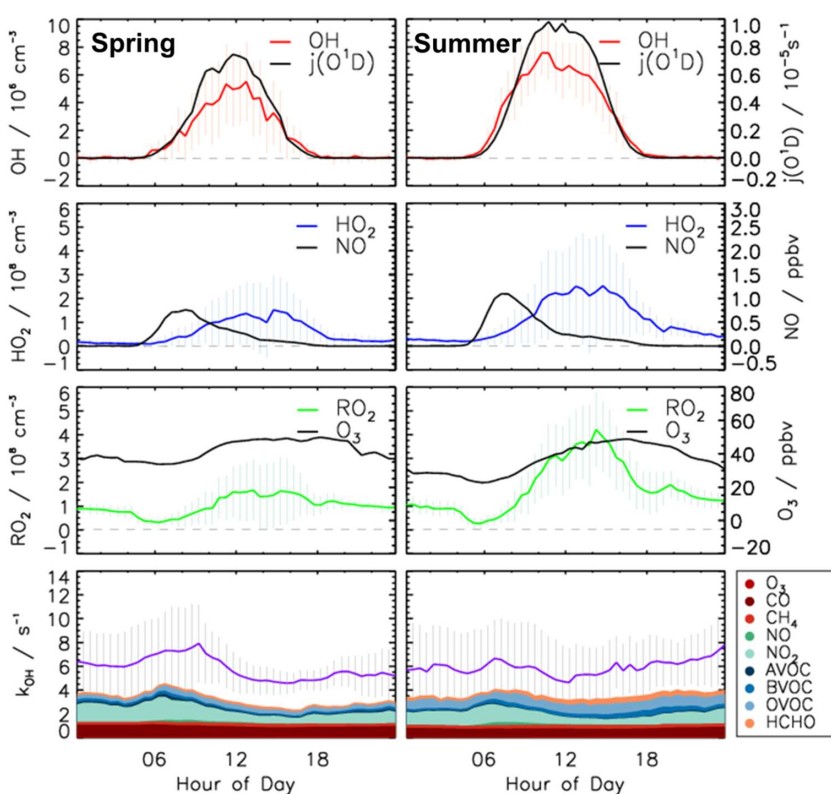

**Figure 5**: Median values of the diurnal profiles of OH, HO$_2$, RO$_2$, $k_{OH}$, $j_{O1D}$, NO, O$_3$ measured in the spring and summer periods of the JULIAC campaign. Colored areas represent the contributions of measured reactants to the total OH reactivity. Vertical lines give 25[th] and 75[th] percentile values.

459

## 3.4 OH, HO$_2$, and RO$_2$ radical concentrations and OH reactivity during the spring and summer periods of the JULIAC campaign

During spring and summer (Fig. 3, 4 and 5), maximum daytime OH concentrations were between 6 and 8 $\times$ 10$^6$ cm$^{-3}$. The highest OH concentration (1.2 $\times$ 10$^7$ cm$^{-3}$) occurred on 31 August. The diurnal OH concentration profile shows a high correlation with the ozone photolysis frequency ($j_{O^1D}$) as observed in previous field campaigns (e.g., Ehhalt and Rohrer (2000); Handisides et al. (2003); Holland et al. (2003)).

Unfortunately, the measurements of HO$_2$ and RO$_2$ radicals were not available for the first two weeks of the spring campaign due to a malfunction of the instrument. Daily maximum HO$_2$ and RO$_2$ concentrations were in the range of 2 to 4 $\times$ 10$^8$ cm$^{-3}$ during the spring period and the first half of the summer period. Maximum HO$_2$ and RO$_2$ concentrations were 8.0 $\times$ 10$^8$ cm$^{-3}$ and 7.0 $\times$ 10$^8$ cm$^{-3}$, respectively, during the second half of summer period. In spring and summer, peroxy radical concentrations showed a distinct diurnal pattern. Both HO$_2$ and RO$_2$ radical concentrations were suppressed in the early morning (between 04:00 and 07:00) due to the reaction with elevated NO mixing ratios of up to 1.5 ppbv. Maximum peroxy radical concentrations were usually reached in the afternoon (~14:00), when NO concentrations were lowest.

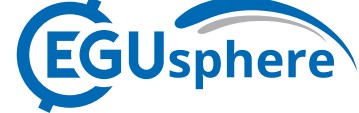

The measured OH reactivity values were in the range of 4 to 18 s$^{-1}$. High values were observed between 23
and 31 August due to high emissions of biogenic volatile organic compounds (BVOCs) from plants at high
ambient temperatures. The OH reactivity that cannot be attributed to the measured OH reactants was on
average, 2.5 s$^{-1}$ (40%), which is much higher than observed in the winter and autumn periods (Fig. S5). CO
and CH$_4$ contributed 10% and 4%, respectively. Due to the high emissions of biogenic organic compounds
in spring and summer, the attributed contribution of organic compounds to the total measured OH reactivity
was 20 % and the contribution of NO$_X$ was only 19 %, much less compared to the winter and autumn
periods. Isoprene had the largest contribution among all VOCs accounting for up to 5 % of the total
measured OH reactivity. Unfortunately, the number of detected VOC species in the JULIAC campaign was
small (Table S2).  This, however, does not impact the analysis in this study as the measured OH reactivity
is used to determine the loss rate of OH radicals.

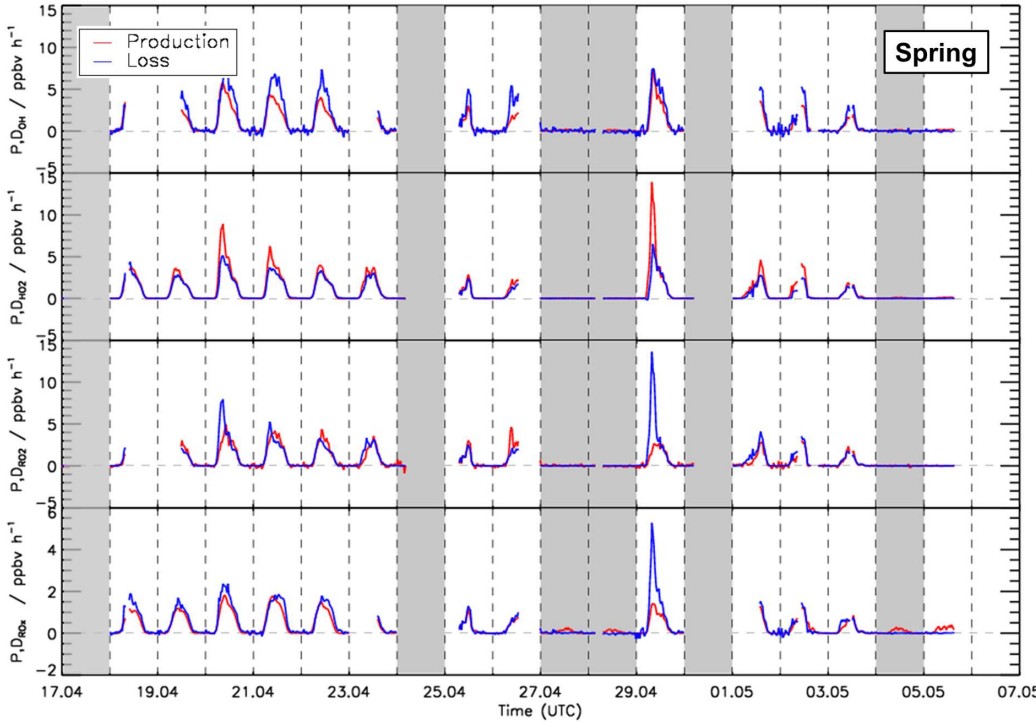

**Figure 6**: Time series of total production and destruction rates of OH, HO$_2$, RO$_2$, and RO$_X$ radicals in the spring period of the JULIAC campaign. Vertical dashed lines denote midnight. Grey areas indicate calibration days and days when the chamber roof was closed.

In the JULIAC campaign, nighttime OH concentrations were clearly below the limit of detection of the
FZJ-CMR-LIF instrument ( $0.7 \times 10^6$ cm$^{-3}$). When all nighttime data are averaged, mean OH
concentrations with 1σ standard errors of $(3 \pm 1) \times 10^4$ cm$^{-3}$ and $(5 \pm 3) \times 10^4$ cm$^{-3}$ are obtained for
the spring and summer periods, respectively. These low values support the absence of instrumentally
produced OH and indicate a very low nocturnal OH production at 50 m height in the absence of NO and
solar UV.



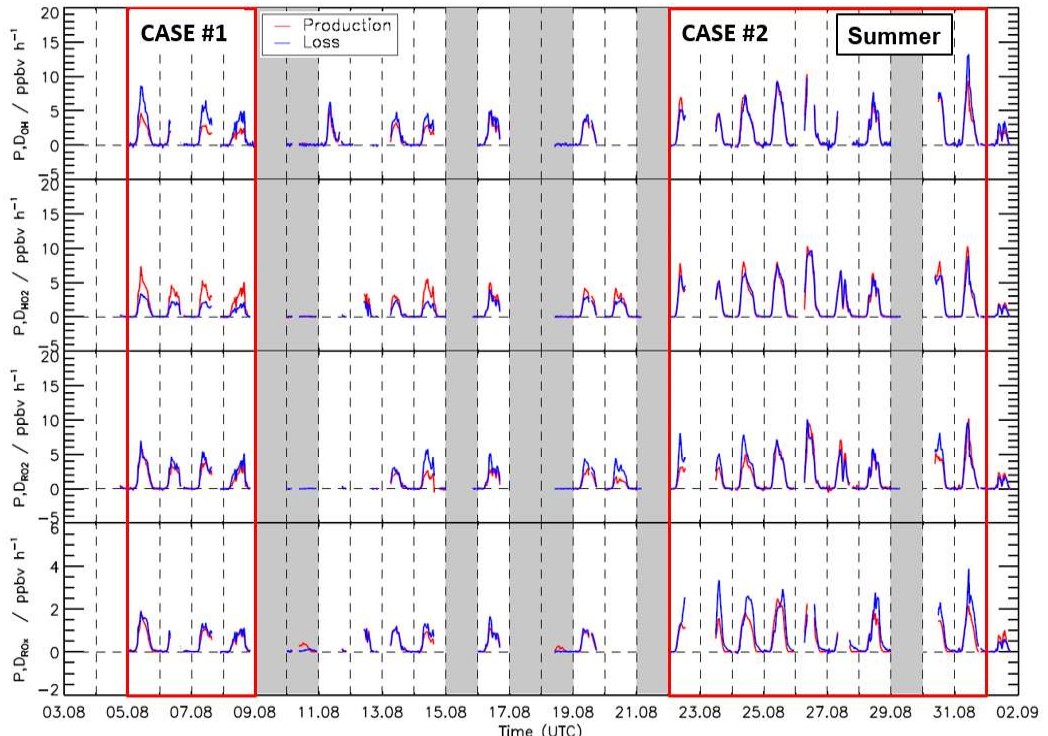

**Figure 7**: Time series of total production and destruction rates of OH, HO$_2$, RO$_2$, and RO$_X$ radicals in the summer period of the JULIAC campaign. Vertical dashed lines denote midnight. Grey areas indicate calibration days and days when the chamber roof was closed. The red boxes denote periods that are discussed in more detail (Case 1 and Case 2).


**3.5 Chemical budgets of OH, HO$_2$, RO$_2$ and RO$_X$ radicals in the spring and summer periods**

Due to the very low photochemical activity observed in autumn and winter, which resulted in radical
concentrations close to the detection limit of the instrument, the chemical budget analysis is only discussed
for data from the spring and summer periods. It focuses on daytime conditions.
Time series of turnover rates of reactions involving OH, HO$_2$, RO$_2$ and RO$_X$ radicals in the spring and
summer periods are presented in Fig. 6 and 7, respectively, and median diurnal profiles in Fig. 8. Typical
daytime turnover rates of OH, HO$_2$ and RO$_2$ radicals were between 3 ppbv h$^{-1}$ and 10 ppbv h$^{-1}$. The rates of
RO$_X$ production and destruction ranged from 1 ppbv hr$^{-1}$ to 3 ppbv hr$^{-1}$, which is 2 to 4 times lower than
those of OH, HO$_2$, and RO$_2$, because radical conversion reactions cancel out. The highest OH turnover rate
of 13 ppbv h$^{-1}$ was observed on 31 August, when the air temperature in the chamber reached up to 40°C.
Unusually high turnover rates for HO$_2$, RO$_2$, and RO$_X$ radicals occurred on 29 April with values of 14 ppbv
h$^{-1}$, 15 ppbv h$^{-1}$, and 4 ppbv h$^{-1}$, respectively, when the NO mixing ratio exceeded 9 ppbv. For the reasons



stated in Section 3.1, the $HO_2$ and $RO_2$ data on this date are considered highly uncertain and were excluded
from further analysis of the chemical budgets.
Diurnal variations of total radical production and destruction rates, as well as of the contributions of the
most important reactions, are shown as median values for the entire spring and summer period in Fig. 8.
For OH, the reaction of $HO_2$ with NO (Reaction R10) was the dominant production pathway contributing
more than 70 % to the total production rate in both spring and summer periods. The photolysis of HONO
(Reaction R1) was the most important primary OH source during daytime contributing approximately 20 %
to the total OH production. The reaction of $HO_2$ with ozone (Reaction R11), the photolysis of ozone
(Reaction R2), and the ozonolysis of alkenes (Reaction R5) contributed less than 3 % to the total OH
production. The maximum median total OH production rate of 3.5 ppbv $hr^{-1}$ was observed in the morning
shortly after the peak NO concentration in both spring and summer (Fig. 5). Values gradually decreased
until sunset. Median total OH destruction rates were higher than production rates and reached up to 5 ppbv
$hr^{-1}$ and 6 ppbv $hr^{-1}$ at noon in spring and summer, respectively. The contributions of different reactions to
the total OH destruction rate is described by the contribution of OH reactants to the OH reactivity (Section
3.4, Fig. 5).
Short-lived radicals are expected to be in a steady state, and therefore radical production and destruction
rates must be balanced. An imbalance between the calculated rates indicates inaccurate data or a missing
radical production or destruction process. The daily peak of the OH production rates was typically lower
than the destruction rate by approximately 1.8 ppbv $h^{-1}$ in the spring and 2.5 ppbv $h^{-1}$ in the summer period
(36 and 43 % of the total OH destruction rate). These discrepancies are higher than the uncertainty of the
calculation (Fig. 8).
80% of the $HO_2$ production rate consisted of the reaction of $RO_2$ with NO (Reaction R9). The remaining
part of the $HO_2$ production rate was due to the photolysis of formaldehyde (9 %) and the reaction of
formaldehyde with OH (10 %). Other reactions producing $HO_2$ played a minor role (< 1 %). The $HO_2$
destruction was mostly due to the reaction of $HO_2$ with NO (Reaction R10) contributing on average 88 %
to the total production rate. The loss due to reaction of $HO_2$ with $RO_2$ radicals (Reaction R16) contributed
on average 9 % to the total loss.
Median values of the total $HO_2$ destruction and production rates were well balanced in the spring period,
with the production rate being slightly higher than the destruction rate. The maximum difference of 1 ppbv
$hr^{-1}$, however, was insignificant compared to the uncertainty of the calculation. A similar tendency but more
pronounced feature was observed in summer. Here, the median value of production rate was higher than
that of the destruction rate by 1.8 ppbv $hr^{-1}$ (38 % of the total $HO_2$ production rate) but differences were
variable (Fig. 7). This aspect is discussed in more detail for two periods (Sections 3.7 and 3.8), which
exhibited different degrees of imbalances in the radical budgets.
The $RO_2$ production rate was dominated by the reaction of VOCs with OH (Reaction R8). The contributions
of ozonolysis of measured alkenes to the $RO_2$ production were very small (less than 1 %). The reaction of
$RO_2$ with NO (Reaction R9) dominated the $RO_2$ destruction and contributed more than 90 % to the total
loss rate. In the late afternoon, the $RO_2$ termination reaction with $HO_2$ gained in importance with
contributions of up to 10 %. Although slight imbalances of up to 1 ppbv were observed in the early morning,
the $RO_2$ production and destruction rates were generally balanced within the uncertainty of calculations in
both spring and summer.



Figure 9 shows the calculated $RO_X$ production and destruction rates. The photolysis of HONO (Reaction
R1), HCHO (Reaction R3) and $O_3$ (Reaction R2) were the dominant processes initiating radical chemistry
and contributed to the total $RO_X$ production rate on average 45 %, 38 % and 15 %, respectively, in both
periods. In the morning, the reaction of OH with $NO_2$ (Reaction R12) was the most important radical
termination process contributing up to 65 % to the total $RO_X$ destruction rate. In addition, due to relatively
high NO mixing ratios in the early morning, the reactions of OH with NO (Reaction R13) and $RO_2$ with
NO, which yields organic nitrate (Reaction R14), were also significant radical termination processes
contributing 13 % and 17 % to the total $RO_X$ destruction rate, respectively. In the afternoon, radical self-
reactions (Reaction R15 – R17), and, in particular, the reaction of $RO_2$ with $HO_2$ (Reaction R16), dominated

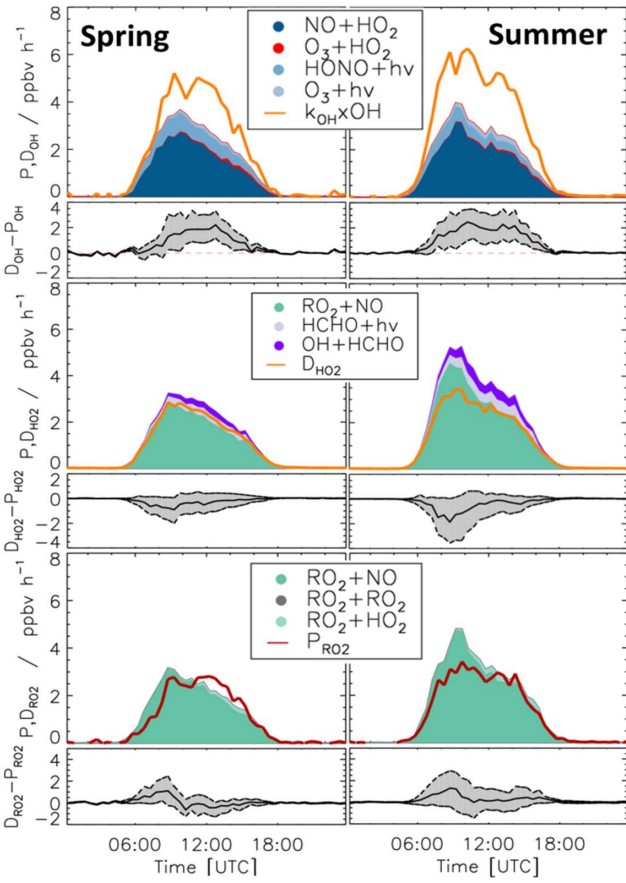

**Figure 8**: Median values of production and destruction rates of OH, $HO_2$, and $RO_2$ radicals in the spring and summer periods of the JULIAC campaign, with data from 29 April excluded. In addition, the differences between the destruction and production rates are shown. Grey areas indicate the uncertainty derived from experimental errors of the measured quantities (Table 2) and of the reaction rate constants (Table 1). The reactions that have insignificant contributions to the production or destruction rates are not shown.





the $RO_X$ destruction due to the low NO and $NO_2$ mixing ratios. In both periods, spring and summer, the
total $RO_X$ destruction rate was slightly higher than the production rate, in particular, in the afternoon. The
imbalance was up to 0.5 ppbv $h^{-1}$, which is higher than the uncertainty of the calculations.
Meteorological and chemical conditions were variable especially in the summer period causing variations
in the balance between radical production and destruction rates (Fig. 7 and Table S3). In the following, the
chemical budgets with the largest and smallest observed imbalances are discussed: August 5-8 (Case 1) and
August 22-31 (Case 2).

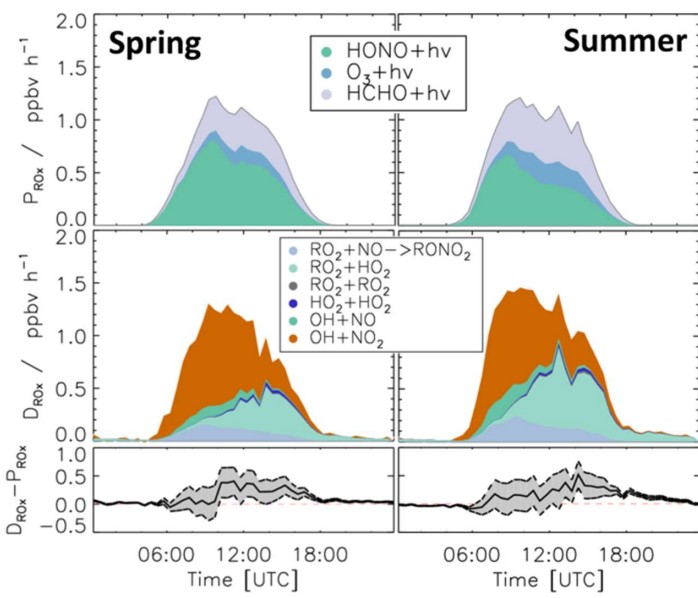

**Figure 9**: Median values of production and destruction rates of $RO_X$ radicals during the spring and summer
periods of the JULIAC campaign. In addition, the differences between the destruction and production rates
are shown. Grey areas indicate the uncertainty derived from experimental errors of the measured quantities
(Table 2) and of the reaction rate constants (Table 1). The reactions that have insignificant contributions
to the production or destruction rates are not shown.

**3.5.1 Case 1: 5 - 8 August 2019**
For the period between 5 and 8 August, relatively low NO mixing ratios (maximum: 1 ppbv, median: 0.26
ppbv) and typical summer temperature for this region (median: 27˚C) were observed (Fig. 10 and Table
S3).
As for the whole summer period (Fig. 8), the reactions of peroxy radicals with NO (Reaction R9, R10)
dominated the inter-radical conversion reactions of OH, $HO_2$ and $RO_2$ in this period (Fig. 10). A significant
imbalance between the OH production and destruction rates of up to 3.0 ppbv $h^{-1}$ (51 % of the total OH
destruction rate) is found, which cannot be explained by the uncertainty of the calculations. The total $HO_2$
production rate was 2.0 ppbv $h^{-1}$ higher than the destruction rate (48 % of the total $HO_2$ production rate),





whilst the RO$_2$ production and destruction rates were well balanced. Relatively small but nevertheless
significant differences between RO$_X$ production and destruction rates (0.5 ppbv h$^{-1}$) were observed during
daytime (Fig. 11).

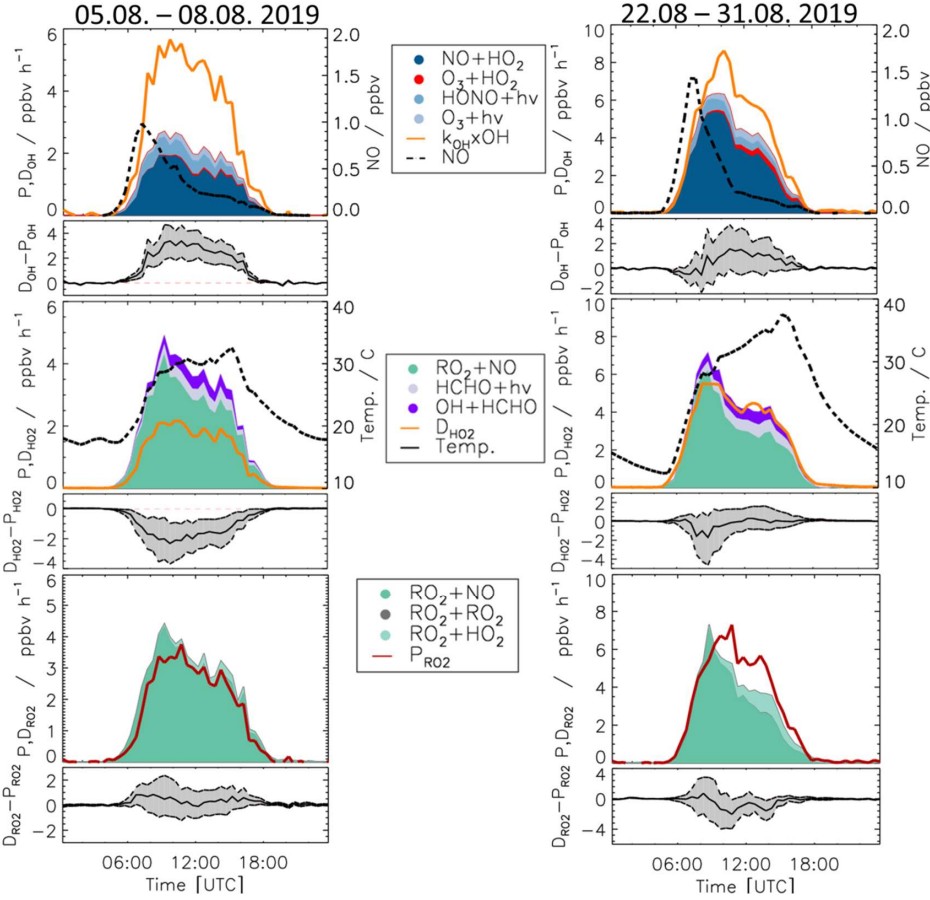

**Figure 10**: Production and destruction rates of OH, HO$_2$, and RO$_2$ radicals for Case 1 (05.08. - 08.08
2019) and Case 2 (22.08 - 31.08 2019). In addition, the differences between the destruction and production
rates are shown. Grey areas give the uncertainty derived from experimental errors of the measured
quantities (Table 2) and of the reaction rate constants (Table 1). The reactions that have insignificant
contributions to the production or destruction rates are not shown.




**3.5.2 Case 2: 22 - 31 August 2019**

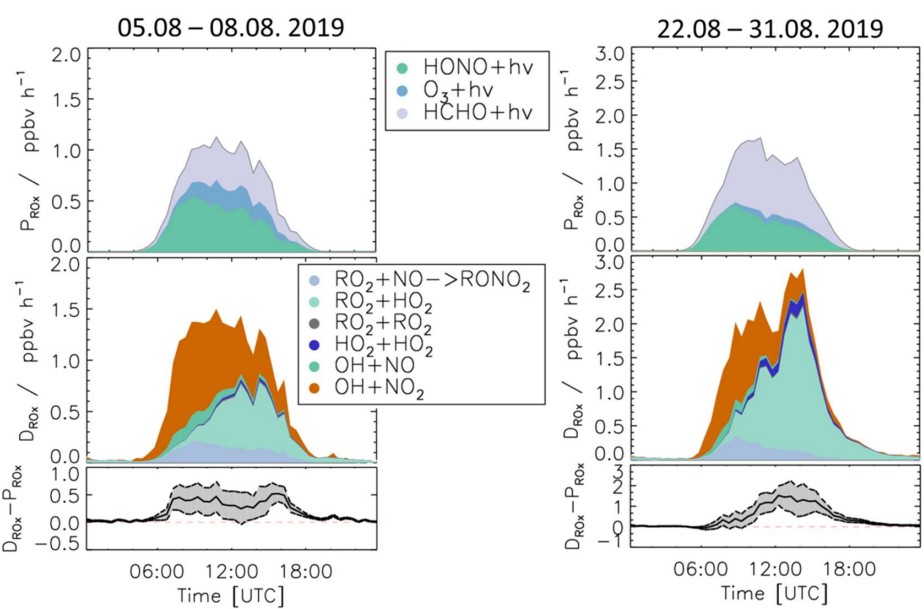

**Figure 11**: Production and destruction rates of $RO_X$ for the periods of the case studies (Case 1 and Case 2). In addition, the differences between the destruction and production rates are shown. Grey areas indicate the uncertainty derived from experimental errors of the measured quantities (Table 2) and of the reaction rate constants (Table 1). The reactions that have insignificant contributions to the production or destruction rates are not shown.

During the period from 22 to 31 August, the temperature was generally high and reached a maximum value
of 42°C inside the chamber. The concentrations of radical precursors, HONO, HCHO and $O_3$, were higher
than those observed in Case 1 (Table S3). Ozone mixing ratios reached values up to 100 ppbv, while
daytime NO mixing ratios were similar as in Case 1 (<1.5 ppbv, median value of 0.22 ppbv). The conditions
outside the chamber were characterized by stagnant air (wind speed < 4 m/s at 50 m height) with no
precipitation. At these conditions, vigorous biogenic emissions can be expected (Vilà-Guerau de Arellano
et al., 2009; Sarkar et al., 2020). Enhanced biogenic VOC emissions and their photochemical degradation
can therefore explain the higher VOC and HCHO concentrations in Case 2 compared to the cooler period
beginning of the month (Table S3). The larger VOC reactivity and comparable OH concentrations resulted
in $HO_2$ and $RO_2$ concentrations that were approximately 2 to 3 times higher than in Case 1 (Table S3).
Imbalances between the radical production and destruction rates were a factor of 2 smaller in the warmer
and more photochemically active period of Case 2 compared to Case 1. OH destruction rates were up to 1.5
ppbv h$^{-1}$ (25 % of the total OH destruction rate) higher than the total production rate (Fig. 10). The $HO_2$
production and destruction rates agree within ±1 ppbv h$^{-1}$. The contributions from photolysis of HCHO and
the reaction of HCHO with OH to the $HO_2$ production rate were larger compared to other periods with





values of up to 15% and 13%, respectively, due to high HCHO mixing ratios of up to 8 ppbv (Fig. 2). The
$RO_2$ production and destruction rates showed imbalances by up to 1.5 ppbv h$^{-1}$ in the late afternoon.
While HONO photolysis was the dominating $RO_X$ source during most of the time in spring and summer
(Fig. 9), $HO_2$ production from the photolysis of HCHO was the most important primary radical source in
Case 2 due to the high concentration of HCHO (Fig. 11). Although the chemical budgets for each radical
species were essentially closed within the experimental uncertainty, the total loss rate of $RO_X$ was
consistently higher than the production rate during daytime. The deviation was higher than the experimental
uncertainty and reached a maximum value of 1.4 ppbv h$^{-1}$ at noontime.

### 3.5.3 NO dependence of radical production and destruction rates

One of the most influential parameters for the radical chemistry is the concentration of NO, since the
reaction with NO dominates the conversion rate of $RO_2$ to $HO_2$ (Reaction R10) and $HO_2$ to OH (Reaction
R9) (Fig. 10). Figure 12 shows the NO dependence of median values of the calculated production and
destruction rates for the different radicals for the spring and summer period.
For OH, the production rates are consistently lower than the destruction rates by about 1.5 ppbv h$^{-1}$ for NO
mixing ratios lower than 1 ppbv NO. At higher NO, the OH budget is balanced within the experimental
uncertainty. For $HO_2$, an inverse pattern is observed. Below 1 ppbv NO, the production rate is higher than
the destruction rate by about 1 ppbv h$^{-1}$. Only for lowest NO mixing ratios, the production and destruction
rates are balanced. For NO mixing ratios above 1 ppbv, the chemical budget of $HO_2$ is essentially closed.
For NO mixing ratios of 3.5 ppbv, the difference between production and destruction rate is noticeably high
with 4 ppbv h$^{-1}$ but has also a large uncertainty. For $RO_2$ radicals, the chemical budget is closed for NO
mixing ratios below 1 ppbv but an increasing discrepancy between the loss and production rates is observed
with increasing NO mixing ratios. While the production rate is relatively constant with a value of 2.5 ppbv
h$^{-1}$, the loss rate increases to values of up to 7.5 ppbv h$^{-1}$ at 3.5 ppbv NO. The budget of $RO_X$, in which
radical inter-conversion reactions cancel out, is mostly balanced over the whole range of NO. Only for
lowest and highest NO mixing ratios the destruction rate is 0.6 ppbv h$^{-1}$ higher than the production rate.





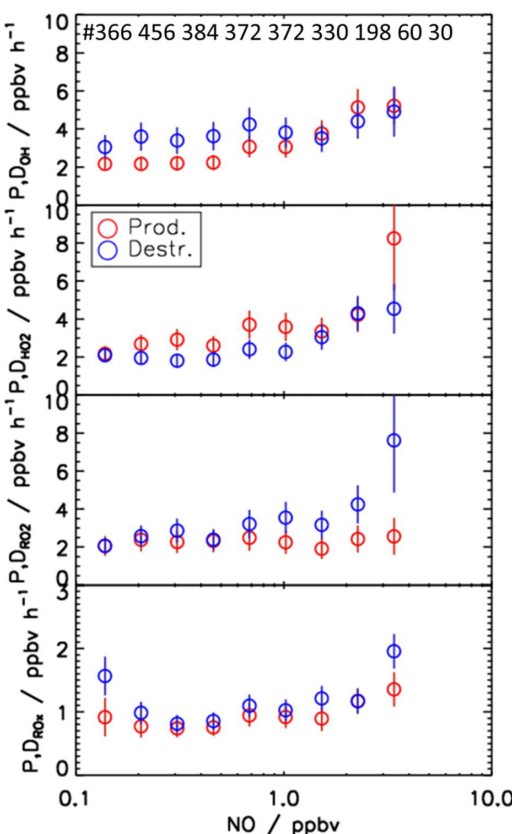

**Figure 12**: NO dependence of median production and destruction rates of OH, $HO_2$, $RO_2$, and $RO_X$ radicals. Median values include all data from the spring and summer periods of the JULIAC campaign (NO intervals 0.4 ppbv). Vertical bars represent the uncertainty from experimental errors of the measured quantities (Table 2) and of the reaction rate constants (Table 1).


**4 Discussion**
**4.1 Discrepancies in the chemical budgets of radicals**
The highest imbalances in the chemical budgets of radicals are found for OH radicals. In spring and summer,
their production rate was consistently lower than the loss rate (Fig. 8). This deficit was largest beginning of
August (Case 1, Fig. 10) when the discrepancy reached $(3.0\pm1)$ ppbv h$^{-1}$.
Imbalances in the radical budgets can be observed for different reasons. They can be caused by missing
processes or incorrect rate constants in the calculations of the production or destruction rates (Section 4.2).
It is also possible that measured concentrations that are used for the calculation contain unknown errors.





The technically difficult radical measurements have a large potential for artefacts (Hofzumahaus and Heard,
2016). Precautions were taken to minimize measurement interferences for OH and $HO_2$ in this campaign:
• The measurements of OH by the LIF instrument were interference-corrected using chemical
modulation and agreed with simultaneous OH measurements by the DOAS instrument within the
experimental uncertainties. The measured OH reactivity quantifies the total chemical loss rate of
OH caused by atmospheric reactants and has a total accuracy of 10%. Thus, the destruction rate of
OH, which is the product of the concentration and reactivity of OH, is known within 20 % and is
unlikely biased by unknown OH interferences or unknown atmospheric reactants.
• The $O_X$ production rate calculated from the reaction of peroxy radicals with NO agrees with the
measured increase of $O_X$ concentrations within $\pm 1$ ppbv h$^{-1}$ for most conditions (Section 3.1). As
more than 70 % of the OH production is due to the reaction of $HO_2$ with NO (Reaction R10), a bias
of more than 1 ppbv h$^{-1}$ due to an unaccounted $HO_2$ measurement error seems unlikely.
• The analysis of the chemical budget of OH in previous chamber experiments performed at various
chemical conditions showed no evidence for a missing OH source originating from chamber wall
effects (Kaminski et al., 2017; Fuchs et al., 2018; Novelli et al., 2018; Rolletter et al., 2019;
Rolletter et al., 2020).
Thus, there is no evidence for instrumental errors that are not included in the estimated errors of the
calculated turnover rates. The observed imbalances in the OH budget of up to 3 ppbv h$^{-1}$ are therefore most
likely due to a missing OH source.
The missing OH production is correlated with the imbalance in the $HO_2$ budget, for which the production
rate is larger than the loss rate at low NO mixing ratios (Fig. 12). This is most clearly seen in the period of
Case 1, when the discrepancy reaches $(2.0\pm 1)$ ppbv h$^{-1}$ (Fig. 10). The production rate of $HO_2$ is nearly equal
to the $RO_2$ loss rate ($P_{HO2} \approx D_{RO2}$) because both are controlled by the reaction of $RO_2$ with NO (Reaction
R9). Furthermore, the $RO_2$ loss rate is well balanced by the $RO_2$ production rate within the experimental
uncertainty of $\pm 1$ ppbv h$^{-1}$ (Fig. 8 and 10). Thus, there is no hint that the calculated turnover rate of the $RO_2$
+ NO reaction had a bias higher than 1 ppbv h$^{-1}$. In addition, turnover rates of the reactions of $HO_2$ and $RO_2$
with NO producing ozone are consistent with the observed $O_X$ increase in the chamber (Section 3.1). This
suggests that these rates are correct in the chemical budget analysis. For the above reasons, the discrepancy
between $HO_2$ production and destruction rates is most likely due to a missing $HO_2$ loss process and not by
measurement errors of $HO_2$, $RO_2$ or NO.
$RO_X$ destruction rates are generally higher than the production rates but differences are on average lower
than 0.5 ppbv h$^{-1}$ (Fig. 9). In the periods of Case 1 and Case 2, the corresponding discrepancies reach 0.5
ppbv h$^{-1}$ and 1.4 ppbv h$^{-1}$, respectively (Fig. 10). If these discrepancies were due to a missing primary OH
source, they could also explain a small part (17 %) of the imbalance in the chemical OH budget in Case 1,
and the complete imbalance in the OH budget in Case 2.
It is difficult to identify the exact cause for the differences in OH and $HO_2$ budgets observed for Case 1 and
2 only with the available data. Case 2 was characterized by high temperature with increased BVOC
emissions and high levels of HCHO (Table S3). No clear correlation was found between the ratio of the
production and destruction rates of the radicals and the concentration of chemical species such as NO, $NO_2$,
$O_3$, HCHO, etc. A weak correlation was observed with temperature with an improved balance in the budgets
the higher the temperature was. This could indicate that the unaccounted processes become less competitive





666 for high radical turnover rates with chemical conditions being dominated by organic compounds from
667 biogenic emissions.

668 In conclusion, the radical budget analysis suggests the presence of a missing OH source and a missing $HO_2$
669 loss process with a similar turnover rate at NO mixing ratios below 1 ppbv for typical temperatures in
670 summer. The opposing imbalances in the OH and $HO_2$ budgets could be due to an unknown mechanism
671 that converts $HO_2$ to OH, or they could indicate a missing primary OH source and a similar fast, but
672 independent termination reaction removing $HO_2$. The remaining imbalance in the $RO_X$ budget would be
673 consistent with an unaccounted primary OH source. This fits best the observations in Case 2 characterized
674 by high temperatures and VOC emissions.

675 For NO mixing ratios that are higher than 1 ppbv, production and destruction rates of OH and $HO_2$ radicals
676 are generally balanced (Fig. 12). An exception is observed for $HO_2$ for highest NO mixing ratios of 3.5
677 ppbv, for which the production rate is 3.5 ppbv h$^{-1}$ higher than the loss rate.

678 For $RO_2$, the radical budget is not closed, but the loss rate increases with NO in contrast to the production
679 rate. The difference reaches a value of 5 ppbv h$^{-1}$ at 3.5 ppbv NO. In the same range of NO mixing ratios,
680 the odd oxygen production rate ($P_{Ox}$) calculated by peroxy radicals (Eq. 14) overestimates the observed
681 increase in the $O_x$ mixing ratio by about 3 ppbv h$^{-1}$. This difference points to a systematic error in the peroxy
682 radical measurements explaining a considerable part of the imbalance in the $RO_2$ budget. A reduction of
683 the $RO_2$ concentration by $3 \times 10^7$ cm$^{-3}$ would reduce the $HO_2$ production rate by 3 ppbv h$^{-1}$ and resolve
684 the discrepancy in the odd oxygen production calculations for the highest NO mixing ratio. The presumed
685 bias in the $RO_2$ measurement may be caused by an incorrect background subtraction that becomes most
686 relevant at high NO concentrations (Section 3.1). However, even after correction of this bias a discrepancy
687 in the $RO_2$ budget would remain requiring an additional $RO_2$ source of approximately 2 ppbv h$^{-1}$ to be
688 balanced.

689 Further information on the nature of the missing $RO_2$ source can be obtained from the chemical budget of
690 $RO_X$, for which the production rate is 0.5 ppbv h$^{-1}$ smaller than the loss rate at 3.5 ppbv NO (Fig. 12). This
691 discrepancy cannot be explained by the instrumental uncertainties in $HO_2$ and $RO_2$ measurements, because
692 the $RO_X$ budget at high NO in the morning was dominated by OH reactions with $NO_2$ and (Fig. 9). Thus,
693 the imbalance in the $RO_X$ budget at high NO indicates a missing primary radical source, which on a single
694 day (29 April) even reached 3 ppbv hr$^{-1}$ (Fig. 6). As the OH budget is balanced for most of the time and
695 the corresponding $HO_2$ budget does not require an additional $HO_2$ source, a missing primary $RO_2$ source is
696 a likely explanation for the discrepancy in the $RO_X$ budget. This would also explain part of the imbalance
697 in the $RO_2$ budget at high NO concentrations.

698

### 699 4.2 Potentially missing chemical processes

700 The above discussion shows that imbalances between calculated production and destruction rates are highly
701 variable over time and change with chemical conditions. As main general features in spring and summer,
702 the radical budget analysis indicates unaccounted OH production processes with a typical strength of 1.5 –
703 3 ppbv h$^{-1}$ at low NO concentrations, which coincides with a missing $HO_2$ sink of 1 – 2 ppbv h$^{-1}$. At high
704 NO mixing ratios (> 1 ppbv), the radical budgets for OH and $HO_2$ radicals are relatively well balanced, but



$RO_2$ production processes of about 2 ppbv h$^{-1}$ appear to be missing in the $RO_2$ radical budget. In the
following, potential reasons for the observed discrepancies in the radical budgets are discussed.

### 4.2.1 Differences in the chemical behavior of specific $RO_2$ radicals

As no speciated $RO_2$ radicals were detected but the sum of all $RO_2$ species, effective rate coefficients for
the reaction of all $RO_2$ species with NO (Reaction R9, R14), $RO_2$ (Reaction R15), and $HO_2$ (Reaction R16)
are used from structure-activity relationship (SAR) by Jenkin et al. (2019) for the calculations of turnover
rates. Potential systematic errors due to this simplification for reactions of $RO_2$ with $RO_2$ and $HO_2$ are
expected to be negligible due to their small contributions to the total turnover rates.
In contrast, the reaction of $RO_2$ with NO plays an important role in the chemical budgets of $HO_2$ and $RO_2$.
The reaction has one channel that converts $RO_2$ to $HO_2$ (Reaction R9) and one radical termination channel
that produces organic nitrates ($RONO_2$) (Reaction R14). The unknown speciation of $RO_2$ causes uncertainty
with respect to the total rate constant of the $RO_2$ + NO reaction ($k_9 + k_{14}$). An effective value of
$9 \times 10^{-12}$ cm$^3$ s$^{-1}$ was taken from (Jenkin et al., 2019). A high limit for the total rate coefficient of $RO_2$
+NO (for example $1.1 \times 10^{-1}$ cm$^{-1}$ s$^{-1}$, 298K for c-$C_5H_9O_2$) would slightly increase the imbalances
between production and destruction rates for $HO_2$ and $RO_2$ radicals by 13 % for both spring and summer.
A lower limit would be the rate constant of the reaction of methyl peroxy radicals ($CH_3O_2$) with NO having
a value of $7.7 \times 10^{-12}$ cm$^{-3}$ s$^{-1}$ (298 K)., Applying this number in the calculations for $HO_2$ production and
$RO_2$ destruction rates (Fig. S6) for the period when observed discrepancies in the $HO_2$ budget were highest
(Case 1) further improves the already well balanced budget of $RO_2$ radicals. This also reduces the imbalance
between $HO_2$ destruction and destruction rates, but the effect is rather small (approximately 10%) and not
sufficient to explain the total difference. For the other periods such as the spring period and the period of
Case 2, a reduced reaction rate would worsen the observed imbalances.
An additional uncertainty in the $HO_2$ production rate comes from the assumed yield of organic nitrates in
the reaction of $RO_2$ with NO. Typical organic nitrate yields range from 5 % to 20 % (Jenkin et al., 2019).
The low limit value is applied in the calculations above. Using a value of 20 % decreases the discrepancy
between $HO_2$ production and destruction rates from 2.0 to 1.5 ppbv h$^{-1}$ for the period of Case 1.
It is worth noting that the organic nitrate yield is generally higher for larger hydrocarbons, but the rate
constant for the $RO_2$ + NO reaction is also often higher, so that there are compensating effects in the
production efficiency of $HO_2$. In addition, it is expected that only a fraction of $RO_2$ radicals is produced
from large hydrocarbons due to the major composition of $RO_2$ would be methyl peroxy radicals.
For the above reasons, the unknown speciation of $RO_2$ is unlikely the reason for the observed imbalances
in the $HO_2$ budget that are most prominent in the period of Case 1.

### 4.2.2 Unaccounted primary radical sources

Primary $RO_X$ radical production that may not be appropriately accounted for in the calculations could be
OH, $HO_2$, and $RO_2$ production from the ozonolysis of alkenes. Only few alkene compounds were measured
in the JULIAC campaign. The contribution from the ozonolysis of these alkenes to the radical production
was very small with values in the range of 0.005 to 0.03 ppbv h$^{-1}$ (Section 3.5). The ozonolysis of small
alkenes such as propene and cis-2-butene that were not measured but are often abundant for example in



forested areas (Goldstein et al., 1996; Rhew et al., 2017), may have significantly contributed to the radical
production.
The potential impact of unmeasured alkenes on the primary radical production is tested by assuming that
the OH reactivity that cannot be explained by measured OH reactants (on average, 2.5 s$^{-1}$) originates from
1.5 ppbv propene and 1.0 ppbv cis-2-butene. The radical production by ozonolysis of the additional propene
and cis-2-butene increases the production from ozonolysis of measured species by more than an order of
magnitude in both spring and summer periods of the JULIAC campaign (Fig. S7) The discrepancies
between the total $RO_X$ production and destruction rates is significantly decreased for the period of the 2
Case studies by approximately 0.2 ppbv h$^{-1}$. However, the additional OH production is by far insufficient
to explain the missing OH source that was generally found during the JULIAC campaign. In addition, the
corresponding OH and $O_3$ reactivity from the additional alkene compounds is about a factor of 6 larger than
of alkenes (e.g., ethene, propene, trans-2-butene, cis-2-pentene) that were measured in ambient air next to
the SAPHIR chamber in the HOxComp campaign in July 2005 (Elshorbany et al., 2012; Kanaya et al.,
756  2012).

The photolysis of oxygenated organic compounds is another source for radicals that could be
underestimated in the calculations. Only the photolysis of HCHO is included in the production rate of $HO_2$
and $RO_X$ at all times of the campaign. In addition, acetaldehyde ($CH_3CHO$), methyl vinyl ketone (MVK),
methacrolein (MACR), and methylglyoxal were measured during part of the campaign and were not
included in the analysis in Section 3. Calculations show that radical production rate from their photolysis
was less than 0.1 ppbv h$^{-1}$. Thus, photolysis of unmeasured OVOCs was very likely unimportant in the
present study. This is consistent with similar small contributions from photolysis of OVOCs other than
HCHO found in in the HOxComp campaign (Kanaya et al., 2012).
The photolysis of $ClNO_2$ constitutes a primary radical source (Reaction R20, R22) that can be found in
coastal environments (e.g., Osthoff et al. (2008)) and mid-continental regions (e.g., Thornton et al. (2010)).
The availability of $ClNO_2$ data during the summer period allowed assessing the potential impact of its
photolysis on the $RO_2$ radical production (Eq. 9). Due to the low mixing ratio of $ClNO_2$ of less than 0.4
ppbv (Tan et al., 2022), the $RO_2$ production from Cl oxidation processes was insignificant (<0.1 ppbv h$^{-1}$)
and cannot explain the observed discrepancies in the primary production and destruction rates of radicals
in the summer period and in the case studies. The instrument detecting $ClNO_2$ was not available in the
spring period of the campaign. Therefore, the extent to which $ClNO_2$ photolysis contributed in spring, for
example to the large missing $RO_x$ source (up to 3 ppbv hr$^{-1}$) on 29 April, remains unknown.
**4.2.3 Unaccounted radical termination reactions**
Heterogeneous uptake of $HO_2$ on aerosol is a potential termination reaction that is not included in the $HO_2$
and $RO_X$ destruction rates above. However, the impact of including the heterogeneous $HO_2$ loss on aerosol
surface (Eq. 8) on the total loss rate is insignificant (less than 1 %), even if a high effective uptake
coefficient of 0.2 is assumed (Fig. S7).
As $HO_2$ uptake is a radical termination process, its relative contribution to the total $RO_X$ loss rate can be
higher compared to the relative contribution to the total $HO_2$ loss rate. However, the only notable influence
would be for the period of Case 2 (8 % of total $RO_X$ loss rate), when the aerosol surface area concentration
was high with values of up to 3.0 $\times 10^2$ $\mu m^2$ cm$^{-3}$.





The estimate for the heterogeneous $HO_2$ loss rate has a high uncertainty because the uptake coefficient
highly depends on the aerosol properties that were not fully characterized in this campaign. Previous
laboratory investigations showed a large variability for the uptake coefficient with values ranging from 0.08
to 0.6 depending on the aerosol chemical composition and the physical state (George et al., 2007; Taketani
et al., 2008, 2009; George et al., 2013; Lakey et al., 2015; Song et al., 2020; Tan et al., 2020). Even the
largest reported $HO_2$ uptake coefficients cannot explain the observed differences in the chemical budget of
$HO_2$ radicals. Therefore, heterogeneous $HO_2$ reactions can be ruled out as an explanation for the
unexplained $HO_2$ loss rate.
**4.2.4 Unaccounted radical inter-conversion reactions**
In the last decade, it has been discovered that unimolecular reactions of $RO_2$ can significantly increase
atmospheric OH concentrations in low-NO environments where they can compete with the reaction of $RO_2$
with NO. The most important, atmospherically relevant example is the production of OH from the
isomerization of isoprene-$RO_2$ radicals (Peeters et al., 2009; da Silva et al., 2010; Peeters and Müller, 2010;
Crounse et al., 2011; Fuchs et al., 2013; Peeters et al., 2014; Teng et al., 2017; Novelli et al., 2020). The
SAPHIR chamber is surrounded by a deciduous forest that emits isoprene especially in summer. Compared
to previous campaigns on the campus where up to several ppbv of isoprene were measured (Komenda et
al., 2003; Spirig et al., 2005; Kanaya et al., 2012), concentrations were relatively low during the JULIAC
campaign (< 0.4 ppbv, on average).
The effect of the conversion of $RO_2$ to OH by the isomerization of isoprene-$RO_2$ (Eq. 4)is tested in the
analysis of the OH and $RO_2$ budgets. In the afternoon of days in the spring period and the period of Case 2,
the total OH production increases only 1 % due to the low isoprene mixing ratios (< 0.2 ppbv) and the
competition of unimolecular reactions with bimolecular reactions of $RO_2$ with NO. Even in the summer
period, when isoprene mixing ratios were up to 0.8 ppbv, the contribution of isomerization reactions from
isoprene-$RO_2$ radicals to the total turnover rate of $RO_2$ is still small with values of less than 4 %. This
implies that unimolecular decomposition reactions of isoprene-$RO_2$ radicals made a minor contribution to
the $RO_2$ destruction and OH production rates.
Another known isomerization process that produces OH applies to $RO_2$ that are formed by OH oxidation
of methacrolein (MACR) (Crounse et al., 2012; Fuchs et al., 2014), which is an oxidation product of
isoprene. MACR mixing ratios were up to 0.5 ppbv in the JULIAC campaign. Because the rate constant for
the OH reaction of MACR is smaller than for isoprene, OH regeneration from MACR-$RO_2$ radicals is even
less important than from isoprene-$RO_2$.
For acyl and carbonyl peroxy radicals it was shown that the reaction of $RO_2$ with $HO_2$, which mainly forms
hydroperoxides (ROOH) (Reaction R16), can produce OH with yields up to 80% (Hasson et al., 2004;
Dillon and Crowley, 2008; Groß et al., 2014; Praske et al., 2015; Winiberg et al., 2016; Jenkin et al., 2019).
It is also noteworthy that the rate constant for the reaction of $HO_2$ with this class of $RO_2$ species is almost
a factor of 2 higher than for other $RO_2$ species (Jenkin et al., 2019). However, even if it is assumed that all
the measured $RO_2$ are acyl and carbonyl peroxy radicals, the formation of OH from their reaction with NO
could only explain up to 0.5 ppbv h$^{-1}$ of the imbalances in both OH and $HO_2$ budgets.



Studies in the remote marine boundary layer show that $HO_2$ to OH conversion mediated by halogen oxides
(XO, X = Cl, Br, I) (e.g., Bloss et al. (2005); Sommariva et al. (2006); Kanaya et al. (2007); Stone et al.
(2018); Fan and Li (2022)) can significantly contribute to the interconversion of radicals and destroy ozone:
$HO_2 + XO \quad \rightarrow \quad HOX + O_2 \quad$ (R23)
$HOX + h\nu \quad \rightarrow \quad OH + X \quad$ (R24)
$XO + NO \quad \rightarrow \quad NO_2 + X \quad$ (R25)
$X + O_3 \quad \rightarrow \quad XO \quad$ (R26)
This conversion mechanism would only be effective at low NO, when the consumption of XO by NO
(Reaction R25) is comparatively slow and when X is not depleted by other reactions as in the case of Cl by
reactions with VOCs (Reaction R22).
For BrO, the rate constants for Reaction R23 and R25 are about the same ($2.1 \times 10^{-11}$ $cm^{-3}\,s^{-1}$ at 298 K,
(Burkholder, 2019). Thus, the reaction of BrO with $HO_2$ would only be dominant, if the NO concentration
were smaller than the concentration of $HO_2$, i.e., less than 10 pptv in this campaign. For IO, the situation is
similar and NO mixing ratios would need to be less than 40 pptv. Such low NO mixing ratios were not
observed during daytime and rule out significant halogen oxide mediated $HO_2$ to OH conversion. The
required XO concentrations to achieve an $HO_2$ loss rate of 1 ppbv $h^{-1}$ at an $HO_2$ concentration of
$2 \times 10^8$ $cm^3$ would be 66 pptv BrO or 16 pptv IO, which exceeds the abundances reported for marine
environments, where halogen sources are known to exist, by more than an order of magnitude. For these
reasons, halogen oxide chemistry cannot explain the missing $HO_2$ sink and missing OH source in this study.
**4.3 Comparison with results from other field campaigns**
Although the chemical and physical conditions were partly influenced by the chamber properties (Section
2.1), the radical concentrations observed during spring and summer were within the range of values that
have been observed in other field studies in summertime in urban and suburban areas (Tan et al., 2001; Ren
et al., 2003; Kanaya et al., 2007; Mao et al., 2010; Lu et al., 2013; Brune et al., 2016; Tan et al., 2017;
Whalley et al., 2018; Tan et al., 2019). The impact of the decreased solar radiation by the chamber
transmission on the radical production was compensated by the radical production from the photolysis of
HONO and HCHO emitted from the chamber film.
This effect is also shown in the relationship between the OH concentration and the photolysis frequencies
of ozone, $j_{O^1D}$ (Section 3.4). The slope ($8.0 \times 10^{11}$ $cm^{-3}\,s^{-1}$) of the correlation for the data from the
JULIAC campaign is much higher than obtained for data in other field campaigns in similar environments
(Ehhalt and Rohrer, 2000; Handisides et al., 2003; Holland et al., 2003; Tan et al., 2017) due to the high
OH production by the photolysis of chamber-produced HONO (Reaction R1). This is further confirmed by
the similarity in OH and $HO_2$ radical concentrations between this campaign and what was observed in the
HOxComp campaign when measurements were performed in front of the SAPHIR chamber for 3 days in
July 2005 (Elshorbany et al., 2012).
In contrast, daytime OH concentrations observed during winter and autumn in the JULIAC campaign were
lower than OH concentrations observed in previous wintertime field campaigns (Heard et al., 2004; Ren et



al., 2006; Kanaya et al., 2007; Tan et al., 2018; Ma et al., 2019). This is due to the lower photolysis frequencies in the chamber compared to outsides, which is not compensated by chamber-produced HONO in wintertime, because the emission strength is low at low temperature and low solar radiation.

Very low nighttime OH concentration in all seasons of the JULIAC campaign (Section 3.4) is consistent with observations in previous field campaigns in rural areas in Germany (Ehhalt and Rohrer, 2000; Handisides et al., 2003; Holland et al., 2003), in which nighttime OH concentrations were less than $1 \times 10^5$ cm$^{-3}$. However, in several other field studies performed in urban areas, nighttime OH concentrations were in the range of 0.2 to $3 \times 10^6$ cm$^{-3}$, for example in China (Lu et al., 2014; Rohrer et al., 2014; Tan et al., 2017; Tan et al., 2018; Ma et al., 2019; Tan et al., 2019; Wang et al., 2019; Whalley et al., 2021), in the US (Martinez et al., 2003; Brune et al., 2016; Griffith et al., 2016), and in the UK (Ren et al., 2003; Vaughan et al., 2012). In these studies, the high nighttime OH concentrations could not be explained by model predictions and raised questions about the presence of potential interferences in nighttime OH signals measured by LIF instruments (Mao et al., 2012; Lu et al., 2014; Novelli et al., 2014).

Similar studies investigating the chemical budgets of OH, HO$_2$, RO$_2$, and RO$_X$ radicals like in this study have been performed for data from field campaigns in a suburban area in the Pearl River Delta (PRD), China, in autumn 2014 (Tan et al., 2019), and in central Beijing, China, (Whalley et al., 2021) in summer 2017.

Tan et al. (2019) observed median values of turnover rates of OH, HO$_2$ and RO$_2$ radicals ranging from 10 to 15 ppbv h$^{-1}$, while rates for RO$_X$ initiation and termination rates were on the order of 3 to 4 ppbv h$^{-1}$ during daytime for chemical conditions affected by anthropogenic emissions. From the comparison between the radical production and destruction rates, a missing OH source and a missing RO$_2$ sink with a similar rate up to 7 ppbv h$^{-1}$ (45 % of the total OH turnover) were found at low NO mixing ratios below 1 ppbv, while HO$_2$ production and destruction rates were balanced. The authors suggested that an additional chemical mechanism is required that efficiently converts RO$_2$ to OH without the involvement of NO. One possibility proposed by Tan et al. (2019) is that HO$_X$ radicals are formed from the auto-oxidation of specific RO$_2$ species which include multifunctional groups such as -OH, -OOH, or -CHO groups.

The analysis of the chemical budget of OH radicals in the JULIAC campaign shows that an unaccounted OH source with a rate ranging between 2 and 3 ppbv h$^{-1}$ (about 50 % of the total OH destruction rate) is required at low NO mixing ratios to balance OH production and destruction rates. This rate is smaller than the rate determined in Tan et al. (2019). However, considering that the OH radical turnover rates in the JULIAC campaign were about half compared to values in the campaign in the PRD area, the relative importance of the unaccounted OH source was comparable in both campaigns. However, the mechanism suggested by Tan et al. (2019) is likely not the only explanation for discrepancies in the radical budgets observed in this study. In the JULIAC campaign, to balance the budget of RO$_2$ radicals rather requires an additional radical source than additional loss processes particularly at high NO mixing ratios above 1 ppbv, and the missing OH sources are likely originating from an HO$_2$ to OH conversion process and/or a missing primary OH source.

Whalley et al. (2021) also investigated the chemical budgets for radicals over a wide range of NO mixing ratios (0.1 to 104 ppbv) from measurement performed in central Beijing, China. Compared to the results in Tan et al. (2019) and to results in this study, the rates of RO$_X$ initiation and termination reactions were 2 to 4 times higher. Also, the rates of radical propagation reactions for OH, HO$_2$ and RO$_2$ radicals were 5 to 10



times higher due to fast inter-radical conversion reactions at conditions with high concentrations of NO.
Similar to the results in this study, an OH source with a high rate of up to 15 ppbv h$^{-1}$ (50 % of the total OH
destruction) was required to balance OH production and destruction rates for low NO mixing ratios. This
unaccounted OH source is more than 3 times higher than that determined in the JULIAC campaign and in
the campaign in China reported by Tan et al. (2019). The HO$_2$ production rate observed in Beijing largely
exceeded the destruction rate by 3 to 5 times for low NO mixing ratios. In contrast, production and
destruction of RO$_2$ and RO$_X$ radicals were well balanced. On the other hand, results for conditions of low
NO concentrations, production and destruction of OH radicals were balanced at high NO mixing ratios,
while very high imbalances of up to 50 ppbv h$^{-1}$ were observed for HO$_2$ and RO$_2$ radicals. Whalley et al.
(2021) showed that reducing the rate constant of the reaction between RO$_2$ and NO by a factor of 10 could
close the gaps between production and destruction rates. The authors suggested that the presence of a
significant fraction of RO$_2$ radicals from the oxidation of large and multifunctional VOCs such as
monoterpenes and long-chain alkanes could explain observations. These radicals can undergo multiple RO$_2$
to RO$_2$ conversion reactions by unimolecular isomerization of alkoxy radicals (RO), which are formed from
the reaction of RO$_2$ with NO, so that no HO$_2$ is produced. Such a RO$_2$ radical reaction chain would be
equivalent to an increased chemical lifetime of RO$_2$ radicals, if RO$_2$ species cannot be distinguished by
instruments like in the sum measurements performed by RO$_X$-LIF instruments. Whalley et al. (2021)
showed that RO$_2$ production by this mechanism would largely reconcile discrepancies between modelled
and measured RO$_2$ concentrations (the model-measurement ratio decreases from 6.2 to 1.8), if the OH
reactivity that could not be accounted for by measured OH reactants is attributed to α-pinene.
Applying a reduced rate constant for RO$_2$ to HO$_2$ propagation reactions as suggested in Whalley et al. (2021)
in the calculations in this study could help explaining the observed discrepancies between HO$_2$ and RO$_2$
production and destruction rates. The largest effect is expected when high NO mixing ratios up to 10 ppbv
like on 29 April is experienced. In this case, a high reduction of the rate constant by a factor of 2 for all
measured RO$_2$ would be required to close the observed gaps between production and destruction rates.
Reduced reaction rate constants of the RO$_2$+NO reaction could be expected for RO$_2$ from large VOCS.
However, the fraction of these RO$_2$ species is expected to be small for conditions of this campaign, even if
OH reactivity that is not explained by measured OH reactants is attributed to large VOCs. Therefore, it
seems unlikely that the mechanism suggested by Whalley et al. (2021) affects the observed discrepancies
in the radical budgets in this study.
It is interesting to point out that similar discrepancies in the OH and HO$_2$ budgets have been observed during
the HOxComp campaign in July 2005 (Elshorbany et al., 2012). Although measurements were only done
for 3 days and despite that these were 14 years earlier than measurements in this work, the chemical
composition was similar with comparable values of NOx, O$_3$, isoprene concentrations and of OH reactivity.
As observed in this study, a missing OH radical source in the range of 2 to 4 ppbv h$^{-1}$ was needed to close
the OH budget for low-NO chemical regimes. The lack of measured RO$_2$ radicals did not allow to perform
a measurement-only budget for HO$_2$ radicals. Nevertheless, model calculations overestimated measured
HO$_2$ radicals after the correction for RO$_2$ radical interferences (Fuchs et al., 2011) by up to 30% at low NO
(Kanaya et al., 2012;Elshorbany et al., 2012). Like in this study, good agreement was found between
modelled and measured OH and HO$_2$ radical concentrations only if an unknown loss process for HO$_2$
radicals that would recycle OH was introduced.
**4.4 Potential role of the missing radical processes on the evaluation of the ozone production rate**





The good agreement of the odd oxygen production rates calculated by the two different methods (Section
3.1) not only gives high confidence in the measured peroxy radical concentrations but also confirms the
current chemical understanding of tropospheric ozone formation from the reaction of peroxy radicals with
NO. Therefore, results demonstrate that accurate predictions of radical concentrations in atmospheric
models are crucial to accurately predict the surface ozone level.
However, the significant level of the missing radical processes found in this study implies the difficulties
in the prediction of the radical concentrations by the models without constraining radicals by their
measurements. In low NO mixing ratios, there are two opposing effects of the missing radical processes on
the $O_3$ formation. At first, a missing OH source and therefore an underestimation of OH concentrations by
the models would lower the loss of $NO_2$ by the reduced reaction rate with OH, and essentially produce more
$O_3$ by its photolysis. Furthermore, the production of $RO_2$ would be under-predicted due to the lower OH
concentrations in the models. At the same time, an unexplained $HO_2$ sink would result in the over-prediction
in $HO_2$ concentrations and thus $O_3$ production. In high NO environments, missing $RO_2$ and $RO_X$ production
processes would result in an underestimation of the $O_3$ production.

**5 Summary and conclusions**
Ambient measurements of atmospheric radicals, trace gases, and aerosol properties were performed during
the Jülich Atmospheric Chemistry Project campaign (JULIAC) using the atmospheric simulation chamber
SAPHIR at Forschungszentrum Jülich, Germany. Ambient air was continuously drawn at a high rate into
the chamber (1 hour residence time) through a 50 m high inlet line for one month in each season throughout
961  2019.

For parts of the campaign, measurements of OH concentrations were achieved by two different methods,
laser-induced fluorescence with a chemical modulation system for zeroing (FZJ-LIF-CMR) and differential
optical absorption spectroscopy (FZJ-DOAS). Measurements of both instruments agreed within 11 % (Cho
et al., 2021).
The production rate of odd oxygen ($O_X$) was determined by using either measured $HO_2$ and $RO_2$
concentrations or $O_3$ and $NO_2$ concentrations measured in the chamber and in the incoming flow. Results
showed excellent agreement between the two different methods confirming that $HO_2$ and $RO_2$ are
responsible for the formation of tropospheric $O_3$ and giving additional confidence in the reliability of peroxy
radical concentration measurements performed in the JULIAC campaign.
An analysis of the chemical budgets of OH, $HO_2$, $RO_2$ and $RO_X$ radicals was performed for data obtained
in the spring and summer periods of the campaign. On average, daytime radical turnover rates ranged
between 3 to 6 ppbv h$^{-1}$ and 4 to 10 ppbv h$^{-1}$ in spring and summer, respectively, for OH, $HO_2$ and $RO_2$
radicals, while total rates of $RO_X$ initiation and termination reactions were below 2.0 ppbv h$^{-1}$. For most
conditions, radical production and destruction rates highly depended on the turnover rate of the reaction of
peroxy radicals with NO. For the total turnover rate of the sum of all radicals ($RO_X$), the photolysis of
HONO and HCHO contributed most to the primary radical production and the reactions of OH with $NO_2$
and $RO_2$ with $HO_2$ dominated the radical termination processes.



Differences between radical production and destruction rates were often small and below the accuracy of
the calculations in the JULIAC campaign in winter and autumn. However, for both spring and summer, an
additional OH source is required to explain the observed discrepancy between production and destruction
rates. The OH production rate of this source would need be on average 2 ppbv h$^{-1}$ and 3 ppbv h$^{-1}$ in the
spring and summer period, respectively. This discrepancy is in the same range as observed for
measurements at the same location during the HOxComp campaign in July 2005 (Elshorbany et al., 2012).
Discrepancies between production and destruction rates of OH radicals were highest for conditions with
low NO mixing ratios in this study. This is similar to findings in other field campaigns in China (Tan et al.,
2017; Tan et al., 2019; Whalley et al., 2021). The high reliability of radical data in this study gives further
confidence that the discrepancies arise from unaccounted chemical processes rather than from instrumental
artefacts.
The highest unaccounted OH source with a rate of 3.0 ppbv h$^{-1}$ (51 % of the observed total OH destruction
rate) is observed in the period from 5 August to 8 August (Case 1), when NO mixing ratios were less than
1 ppbv and median maximum temperature in the chamber were 31° C. At the same time, an additional
HO$_2$ destruction process with a rate of up to 2.0 ppbv h$^{-1}$ is required to balance the HO$_2$ production rate,
while production and destruction rates for RO$_2$ radicals are well balanced. This indicates that an
unaccounted HO$_2$ to OH radical propagation process could be present. In addition, part of the missing OH
source could have been originated from a missing primary OH production process, because also a small
difference between the total RO$_X$ production and destruction rates are observed. The missing RO$_X$ source
was up to 0.5 ppbv h$^{-1}$ for Case 1, but was even higher with a rate of 1.4 ppbv h$^{-1}$ in the summer, when
temperature were highest (Case 2).
For NO mixing ratios in range of 1 to 3 ppbv, production and destruction rates for OH and HO$_2$ radicals
were balanced, while additional sources of RO$_2$ and RO$_X$ having on average rates of 1.6 ppbv h$^{-1}$ and 0.4
ppbv h$^{-1}$, respectively, were required to balance their production and destruction rates. Therefore, part of
the missing RO$_2$ source can be explained by a primary radical source, but the remaining RO$_2$ source is still
unresolved.
For high NO mixing ratios above 3 ppbv, 4 to 5 ppbv h$^{-1}$, large discrepancies between production and
destruction rates of HO$_2$ and RO$_2$ radicals were found, but the calculations for these conditions have a higher
uncertainty due to low HO$_2$ and RO$_2$ concentrations close to background signals. Whereas the imbalance in
the budget for HO$_2$ radicals is due to an unaccounted loss processes, an additional RO$_2$ production processes
is required to close the chemical budget for RO$_2$ radicals. For the same conditions, a primary RO$_X$ source
with a rate of 0.5 ppbv h$^{-1}$ was needed to balance the RO$_X$ destruction rate. Therefore, the missing primary
RO$_X$ source is likely an unaccounted primary RO$_2$ source.
Production of radicals from the oxidation of organic compounds by chlorine could have been one additional
source. Unfortunately, the potential impact of chlorine chemistry could not be examined in the spring
periods, when these conditions were experienced, because ClNO$_2$ measurements were not available. During
times when ClNO$_2$ concentrations were measured, chlorine chemistry initiated by the photolysis of ClNO$_2$
did not significantly contribute to the radical production.
For chemical conditions when the contribution of the reaction of HO$_2$ with NO to the OH production was
reduced, i.e. at lower NO levels, other radical formation pathways such as isomerization reactions of RO$_2$



radicals, OH formation from ozonolysis of alkenes or photolysis of multifunctional organic compounds
could gain in importance and need to be properly accounted for. These processes remain relatively poorly
constrained due to the lack of direct measurements of e.g., multifunctional organic compounds.
Although the exact mechanism for the missing production or destruction processes for OH, $HO_2$ and $RO_2$
radicals could not be determined from measurements in this campaign, knowing the magnitudes of the
missing radical processes gives indicative information about the disagreements of model simulations and
observations for radicals and secondary air pollutants.
More investigations of the chemical budgets of radicals for example in environments with high NO mixing
ratios including the determination of the impact of chlorine chemistry and with a detailed characterization
of the chemical composition of air masses with respect to the presence of complex organic compounds
would be beneficial for the understanding of radical chemistry as well as of the formation of secondary air
pollution such as ozone.

**Code and data availability**

Data of the JULIAC campaign analyzed in this work is available from the Jülich Data repository
(https://doi.org/10.26165/JUELICH-DATA/3J80BW, Cho et al., 2022).

**Author contributions**

AH designed JULIAC campaign and organized it together with HF and FH. CC performed the
measurements of radicals, analyzed the data, and wrote the paper together with AN and HF. All co-authors
contributed with data and helped the writing by intensive discussions of the manuscript.

**Competing interests**

The authors declare that they have no conflict of interest.

**Financial Support**

This project has received funding from the European Research Council (ERC) under the European Union's
Horizon 2020 research and innovation program (SARLEP grant agreement no. 681529) and from the
European Commission (EC) (Eurochamp 2020 project, grant agreement no. 730997).

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
