# Peer review of "Experimental chemical budgets of OH, HO2 and RO2 radicals in rural air in West-Germany during the JULIAC campaign 2019"

_EGUsphere, 2022_

## Author Comment (AC2)

**Response to the reviewers for the* "Experimental chemical budgets of OH, HO2 and RO2 radicals in rural air in West-Germany during the JULIAC campaign 2019" study *by* Changmin Cho et al.**

We would like to thank the reviewers for careful reading the paper and providing useful comments which improved the manuscript. We reply here to each comment in blue. Red represents the corresponding correction in manuscript.

**Anonymous Referee #1**

**Comment #1:** P8 L204-207: It is stated that the ROx-LIF system is calibrated for  $CH_3O_2$ . Since the sensitivity of this instrument is species dependent, this will lead to a measurement bias for  $RO_2$ . By how much could the measured  $RO_2$  deviate from the true value? Is this bias factored in the measurement uncertainty? If not, how could it affect the calculations of  $P_{OH,isop}$ , (Eq. 4),  $P_{HO2}$  (Eq. 6),  $D_{HO2}$  (Eq. 7),  $D_{RO2}$  (Eq. 11) and  $D_{ROX}$  (Eq. 13)?

**Response:** As the reviewer pointed out, the detection sensitivity of specific RO2 radicals can be different from the sensitivity of CH3O2. As described in Fuchs et al. (2008), ROxLIF sensitivities were a maximum of 10% lower for RO2 from C1-C3 alkanes and monoalkenes compared to CH3O2, slightly higher for RO2 from isoprene (+20%), and 40% lower for RO2 from isobutane. Recent tests performed with a series of VOC including monoterpenes (limonene, myrcene,  $\Delta$ -3-carene) as well as hexane and isobutane revealed that, for the NO and CO mixing ratios applied in the ROx converter during the JULIAC campaign, a sensitivity within 15% of the sensitivity for CH3O2 could be observed. The conditions in the ROx converter were indeed optimized to reduce any sensitivity dependency on a specific RO2. As this value is within the accuracy of the RO2 measurement it would not affect the analysis performed where the differences observed are much larger than the 15% possible bias in sensitivity.

The following text was added to the manuscript.

'The resulting calibration is also applicable to the majority of other atmospheric alkyl peroxy radicals (Fuchs et al., 2008; Fuchs et al., 2011) and recent laboratory tests performed with a variety of VOCs including monoterpenes and chained alkanes for the CO and NO mixing ratios applied in the ROx converter during the JULIAC campaign showed a decrease of less than 15% of sensitivity as compared to methyl peroxy radicals which is within the accuracy of the instrument.'

**Comment #2:** P10 L306: "First, the contributions from CO, NO, NO2, HCHO and O3 is removed from the measured OH reactivity as these species do not form RO2 radicals in the reaction with OH. It is then assumed that the remaining fraction can be attributed to organic compounds (VOC reactivity ( $k_{VOC}$ )) including measured and unmeasured VOCs, which produce RO2 radicals in their reaction with OH" – For some VOCs the reaction with OH can lead to the prompt formation of HO2 together with RO2. For instance, toluene+OH will form 28% HO2 and 72% RO2. Assuming only the formation of RO2 could lead to an

underestimation of  $PHO_2$  and an overestimation of  $PRO_2$ . Could the authors comment on this aspect? Can the prompt formation of  $HO_2$ , which occurs with a few VOCs, be neglected when the total pool of VOCs is considered?

**Response:** The reviewer raises a good point and we investigated the potential impact of prompt formation of HO2 for RO2 radicals from some VOCs on the radical budgets. In this campaign, toluene, benzene, xylene, phenol and cresol were measured and RO2 from OH addition to these species form prompt HO2 (Nehr et al., 2011; Nehr et al., 2014; Jenkin et al., 2019). However, during JULIAC, their concentrations were small and their average contributions to the OH reactivity from VOCs ( $k_{VOC}$ ) only 2.8%. Therefore, the potential impacts on both, the production rate of HO2 and loss rate of RO2 is negligible (less than 1% of the total rates). With this, we extended the Section 2.3.3.

**Comment #3:** Table 2: This table indicates that  $NO_2$  was measured using a chemiluminescence instrument. Was this instrument equipped with a photolytic  $NO_2$  converter or a molybdenum converter? This should be clearly stated. Instruments equipped with a molybdenum converter are known to be prone to interferences when measuring  $NO_2$ . If a molybdenum converter was used, the authors should discuss how interferences on  $NO_2$  measurements could impact the calculations of ROx destruction rates (Eq. 13) and Ox production rates (Eq. 14).

**Response:**  $NO_2$  was measured using a chemiluminescence instrument with a custom-built photolytic converter. We add a note to Table 2 'NO2 was converted to NO using a custom-built photolytic converter before detection.' and modify P8 L229 'NO and NO2 were measured by chemiluminescence 230 (Eco Physics, TR780, NO2 conversion by a custom-built photolytic converter)'.

**Minor comments:**

**Comment #4:** P8 L226-228: "Photolysis frequencies inside the chamber were derived from the solar actinic flux densities measured by a spectroradiometer mounted on the roof of the nearby institute building (Bohn et al., 2005; Bohn and Zilken, 2005)." – How is the Teflon sheet transmission determined when the cleanliness changes from day-to-day?

**Response:**

A similar question is raised in Comment #10 by Referee #2 and is answered together here. The solar actinic flux inside the chamber is different from the radiation outside due to shadowing by the construction elements of the chamber and due to the transmission of the teflon foil. The differences were derived by a specific radiative transfer model approach. Spectra of the solar actinic flux measured on the roof of a nearby building and a chamber specific model were used to calculate the actinic flux and photolysis frequencies inside the chamber (Bohn et al., 2005; Bohn and Zilken, 2005). The Teflon film cleanliness was no subject to day-to-day changes. Foil transmission normally changes slowly within years. Changes of the transmission of the teflon foil over time are regularly calibrated against chemical actinometer experiments

in which  $NO_2$  is photolyzed in synthetic air in the chamber (Bohn et al., 2005). The actinometer experiments were performed at the beginning and end of each intensive phase during JULIAC. A reported accuracy of 18% was achieved and included in the error analysis for the radical budget.

To clarify this, we modify P7 L162 to be 'Reference experiments with clean synthetic air were performed to investigate possible changes in the strength of chamber emissions and to check for instrumental backgrounds. In addition, chemical actinometer experiments were performed in the chamber, in which  $NO_2$  was photolyzed in synthetic air, before and after each intensive period. The comparison of actinometric and spectroradiometric  $j_{NO2}$  values was used to track and correct for changes in light transmission due to aging of the chamber wall (Bohn et al., 2005).'.

In addition, we change P8 L227 to be 'Photolysis frequencies inside the chamber were derived from solar actinic flux densities measured by a spectroradiometer mounted on the roof of the nearby institute building. Chamber values were calculated using a model approach considering shading effects and the influence of the chamber film (Bohn et al., 2005; Bohn and Zilken, 2005).'

**Comment #5:** P11 L338-339: Was the humidity dependence of k17 accounted for in the calculation of DROx? It seems so from Table 1 where k17 is reported for 1% water but it should be clearly stated in the text.

**Response:** Yes, the water vapour dependence was always considered in the calculations. This is now explicitly stated in the text.

**Comment #6:** P12 L370-371: Please include O3+alkenes in the list of minor Ox destruction pathways

**Response:** Corrected. The modified sentence reads now 'This calculation neglects minor  $O_X$  destruction processes such as the reaction of  $O_3$  with NO2, OH, HO2, Cl or alkenes since they did not play a notable role during the day in this campaign.'

**Comment #7:** Figures 3, 4, S3, S4: Please indicate in the caption what the error bars represent for OH. Also please add error bars for the other measurements.

**Response:** We add on the captions 'The vertical bars represent  $1\sigma$  statistical errors.'

**Comment #8:** Figures 8-12 & S6-S7: Please clarify in the caption whether uncertainties are displayed as 1-sigma? 2 sigma? Other?

Response: Corrected.

**Anonymous Referee #2**

**Comment #9:** In the abstract (lines 39-41), the authors state that the missing OH source "consists likely of a combination of a missing primary radical source ( $0.5 \sim 1.4 \text{ ppbv h}^{-1}$ ) and a missing inter-radical HO2 to OH conversion reaction with a rate of up to 2.5 ppbv h-1." However, there appears to be little discussion of this potential OH source/HO2 sink in the paper, except briefly on page 29 (lines 670-671) and page 34 (line 893) and it is not mentioned in the conclusions. If this is a major finding as suggested in the abstract, it should be emphasized more in the manuscript.

Response: The chemical mechanisms behind the inter-radical  $HO_2$  to OH conversion independent from NO remains difficult to identify. A potential path driven by halogen oxides is discussed in section 4.2.4 although the conclusion is that the concentration of halogen oxides needed to explain the discrepancy is not realistic for the JULIAC campaign. Although it is mentioned in the conclusion (P37 L994-995), we agree that it was not emphasized as the review's comment. We extended the conclusion in the newer version.

**Comment #10:** The authors state that photolysis frequencies were "were derived from the solar actinic flux densities measured by a spectroradiometer mounted on the roof of the nearby institute building." Given that an underestimation of radical production from photolysis could account for the missing OH radical source, the authors should clarify how potential differences in photolysis rates inside versus outside of the chamber were accounted for in their budget calculations.

**Response:** See our response to Comment #4 by Referee #1 who asked a similar question.

**Comment #11:** The authors measured total OH reactivity and use it to determine the total OH loss rate. However, as illustrated in Figure 5 there appears to be significant missing OH reactivity when compared to the calculated reactivity from measured OH sinks. Unfortunately, there is little discussion about the potential composition of the missing OH reactivity. The paper would benefit from a brief discussion of the missing OH reactivity and whether unmeasured OVOCs may be responsible. While the authors suggest that OVOCs such as acetaldehyde, methyl vinyl ketone, methacrolein, and methylglyoxal do not contribute significantly to radical production, have the authors considered other potential unmeasured OVOCs, perhaps through a model of the chemistry, that may be contributing to the missing reactivity as well as be a potential unmeasured radical source?

**Response:** Modelled OVOCs can often help close the gap between measured and modelled OH reactivity. Model simulation on the  $k_{OH}$  were performed in the studies by Kanaya et al. (2012) and by Elshorbany et al. (2012) both analyzing the data collected during the HOxCOMP campaign at the same site as the JULIAC

campaign. The study by Kanaya et al. (2012) tried to match the measured reactivity by increasing primary emissions while the study by Elshorbany et al. (2012) showed an increase of OH reactivity due to oxygenated secondary products (in particular, from isoprene oxidation). Although the measured  $k_{OH}$ , which was in the range as measured during the JULIAC campaign, could be matched by the modelled one in both studies, similar discrepancies in the OH and HO2 radical budgets remained unless an unknown loss process for HO2 radicals that would recycle OH was introduced. If the missing OH reactivity (~ 2.5 s-1) would be all due to glyoxal, an additional OH production of 0.3 ppbv h-1 could be expected which would not be enough to explain the missing radical source observed similar to what was found for the HOxCOMP campaign.

We added 'During the HOxCOMP campaign performed in 2005 at the same site as the JULIAC campaign, the modelled OH reactivity could be matched with the measured reactivity by including either additional primary emissions (Kanaya et al., 2012) or model-produced oxygenated secondary products (Elshorbany et al., 2012). Neither of the additional species contributed enough to close the radical budgets. If it is assumed that the missing OH reactivity (2.5 s-1) is all due to glyoxal (9 ppb) an additional OH production of 0,3 ppb h-1 could be expected. This would still not be enough to close the radical budget suggesting that unmeasured OVOCs do not play a large role.' on P19 L483.

**Comment #12:** P2, L63: The Griffith et al. 2016 reference reports urban measurements. Did the authors mean to cite Griffith et al., Atmos. Chem. Phys., 13, 5403–5423, 2013, which reports measurements in a forest environment?

**Response: Corrected.**

**Comment #13:** In Figure 12 I assume that the numbers at the top of the figure represent the number of points in each NO bin. This should be clarified in the caption. Also, the uncertainty should be clarified.

Response: Done.

**References**

Bohn, B., Rohrer, F., Brauers, T., and Wahner, A.: Actinometric measurements of  $NO_2$  photolysis frequencies in the atmosphere simulation chamber SAPHIR, Atmos. Chem. Phys., 5, 493-503, doi:10.5194/acp-5-493-2005, 2005.

Bohn, B., and Zilken, H.: Model-aided radiometric determination of photolysis frequencies in a sunlit atmosphere simulation chamber, Atmos. Chem. Phys., 5, 191-206, doi:10.5194/acp-5-191-2005, 2005.

Elshorbany, Y. F., Kleffmann, J., Hofzumahaus, A., Kurtenbach, R., Wiesen, P., Brauers, T., Bohn, B., Dorn, H.-P., Fuchs, H., Holland, F., Rohrer, F., Tillmann, R., Wegener, R., Wahner, A., Kanaya, Y., Yoshino, A., Nishida, S., Kajii, Y., Martinez, M., Kubistin, D., Harder, H., Lelieveld, J., Elste, T., Plass-Dülmer, C., Stange, G., Berresheim, H., and Schurath, U.: HOx budgets during HOxComp: A case study of HOx chemistry under NOx-limited conditions, J. Geophys. Res.: Atmos., 117, doi:10.1029/2011JD017008, 2012.

Fuchs, H., Holland, F., and Hofzumahaus, A.: Measurement of tropospheric RO2 and HO2 radicals by a laser-induced fluorescence instrument, Rev. Sci. Instrum., 79, 084104, doi:10.1063/1.2968712, 2008.

Jenkin, M. E., Valorso, R., Aumont, B., and Rickard, A. R.: Estimation of rate coefficients and branching ratios for reactions of organic peroxy radicals for use in automated mechanism construction, Atmos. Chem. Phys., 19, 7691-7717, doi:10.5194/acp-19-7691-2019, 2019.

Kanaya, Y., Hofzumahaus, A., Dorn, H. P., Brauers, T., Fuchs, H., Holland, F., Rohrer, F., Bohn, B., Tillmann, R., Wegener, R., Wahner, A., Kajii, Y., Miyamoto, K., Nishida, S., Watanabe, K., Yoshino, A., Kubistin, D., Martinez, M., Rudolf, M., Harder, H., Berresheim, H., Elste, T., Plass-Dülmer, C., Stange, G., Kleffmann, J., Elshorbany, Y., and Schurath, U.: Comparisons of observed and modeled OH and HO2 concentrations during the ambient measurement period of the HOxComp field campaign, Atmos. Chem. Phys., 12, 2567-2585, doi:10.5194/acp-12-2567-2012, 2012.

Nehr, S., Bohn, B., Fuchs, H., Hofzumahaus, A., and Wahner, A.: HO2 formation from the OH + benzene reaction in the presence of O2, Physical Chemistry Chemical Physics, 13, 10699-10708, doi:10.1039/C1CP20334G, 2011.

Nehr, S., Bohn, B., Dorn, H. P., Fuchs, H., Häseler, R., Hofzumahaus, A., Li, X., Rohrer, F., Tillmann, R., and Wahner, A.: Atmospheric photochemistry of aromatic hydrocarbons: OH budgets during SAPHIR chamber experiments, Atmos. Chem. Phys., 14, 6941-6952, doi:10.5194/acp-14-6941-2014, 2014.

---

## Author Comment (AC3)

*Response to the reviewers for the* "Experimental chemical budgets of OH, HO$_2$ and RO$_2$ radicals in rural air in West-Germany during the JULIAC campaign 2019" study *by* Changmin Cho et al.

We would like to thank the reviewers for careful reading the paper and providing useful comments which improved the manuscript. We reply here to each comment in blue. Red represents the corresponding correction in manuscript.

**Anonymous Referee #1**

**Comment #1:** P8 L204-207: It is stated that the ROx-LIF system is calibrated for CH$_3$O$_2$. Since the sensitivity of this instrument is species dependent, this will lead to a measurement bias for RO$_2$. By how much could the measured RO$_2$ deviate from the true value? Is this bias factored in the measurement uncertainty? If not, how could it affect the calculations of P$_{OH,isop}$, (Eq. 4), P$_{HO2}$ (Eq. 6), D$_{HO2}$ (Eq. 7), D$_{RO2}$ (Eq. 11) and D$_{ROx}$ (Eq. 13)?

**Response:** As the reviewer pointed out, the detection sensitivity of specific RO$_2$ radicals can be different from the sensitivity of CH$_3$O$_2$. As described in Fuchs et al. (2008), RO$_X$LIF sensitivities were a maximum of 10% lower for RO$_2$ from C1-C3 alkanes and monoalkenes compared to CH$_3$O$_2$, slightly higher for RO$_2$ from isoprene (+20%), and 40% lower for RO$_2$ from isobutane. Recent tests performed with a series of VOC including monoterpenes (limonene, myrcene, Δ-3-carene) as well as hexane and isobutane revealed that, for the NO and CO mixing ratios applied in the RO$_X$ converter during the JULIAC campaign, a sensitivity within 15% of the sensitivity for CH$_3$O$_2$ could be observed. The conditions in the RO$_X$ converter were indeed optimized to reduce any sensitivity dependency on a specific RO$_2$. As this value is within the accuracy of the RO$_2$ measurement it would not affect the analysis performed where the differences observed are much larger than the 15% possible bias in sensitivity.

The following text was added to the manuscript.

'The resulting calibration is also applicable to the majority of other atmospheric alkyl peroxy radicals (Fuchs et al., 2008; Fuchs et al., 2011) and recent laboratory tests performed with a variety of VOCs including monoterpenes and chained alkanes for the CO and NO mixing ratios applied in the RO$_X$ converter during the JULIAC campaign showed a decrease of less than 15% of sensitivity as compared to methyl peroxy radicals which is within the accuracy of the instrument.'

**Comment #2:** P10 L306: "First, the contributions from CO, NO, NO$_2$, HCHO and O$_3$ is removed from the measured OH reactivity as these species do not form RO$_2$ radicals in the reaction with OH. It is then assumed that the remaining fraction can be attributed to organic compounds (VOC reactivity (k$_{VOC}$)) including measured and unmeasured VOCs, which produce RO$_2$ radicals in their reaction with OH" – For some VOCs the reaction with OH can lead to the prompt formation of HO$_2$ together with RO$_2$. For instance, toluene+OH will form 28% HO$_2$ and 72% RO$_2$. Assuming only the formation of RO$_2$ could lead to an

underestimation of $PHO_2$ and an overestimation of $PRO_2$. Could the authors comment on this aspect? Can the prompt formation of $HO_2$, which occurs with a few VOCs, be neglected when the total pool of VOCs is considered?

**Response:** The reviewer raises a good point and we investigated the potential impact of prompt formation of $HO_2$ for $RO_2$ radicals from some VOCs on the radical budgets. In this campaign, toluene, benzene, xylene, phenol and cresol were measured and $RO_2$ from OH addition to these species form prompt $HO_2$ (Nehr et al., 2011; Nehr et al., 2014; Jenkin et al., 2019). However, during JULIAC, their concentrations were small and their average contributions to the OH reactivity from VOCs ($k_{VOC}$) only 2.8%. Therefore, the potential impacts on both, the production rate of $HO_2$ and loss rate of $RO_2$ is negligible (less than 1% of the total rates). With this, we extended the Section 2.3.3.

**Comment #3:** Table 2: This table indicates that $NO_2$ was measured using a chemiluminescence instrument. Was this instrument equipped with a photolytic $NO_2$ converter or a molybdenum converter? This should be clearly stated. Instruments equipped with a molybdenum converter are known to be prone to interferences when measuring $NO_2$. If a molybdenum converter was used, the authors should discuss how interferences on $NO_2$ measurements could impact the calculations of ROx destruction rates (Eq. 13) and Ox production rates (Eq. 14).

**Response:** $NO_2$ was measured using a chemiluminescence instrument with a custom-built photolytic converter. We add a note to Table 2 '$NO_2$ was converted to NO using a custom-built photolytic converter before detection.' and modify P8 L229 'NO and $NO_2$ were measured by chemiluminescence 230 (Eco Physics, TR780, $NO_2$ conversion by a custom-built photolytic converter)'.

**Minor comments:**

**Comment #4:** P8 L226-228: *"Photolysis frequencies inside the chamber were derived from the solar actinic flux densities measured by a spectroradiometer mounted on the roof of the nearby institute building (Bohn et al., 2005; Bohn and Zilken, 2005)."* – How is the Teflon sheet transmission determined when the cleanliness changes from day-to-day?

**Response:**

A similar question is raised in Comment #10 by Referee #2 and is answered together here. The solar actinic flux inside the chamber is different from the radiation outside due to shadowing by the construction elements of the chamber and due to the transmission of the teflon foil. The differences were derived by a specific radiative transfer model approach. Spectra of the solar actinic flux measured on the roof of a nearby building and a chamber specific model were used to calculate the actinic flux and photolysis frequencies inside the chamber (Bohn et al., 2005; Bohn and Zilken, 2005). The Teflon film cleanliness was no subject to day-to-day changes. Foil transmission normally changes slowly within years. Changes of the transmission of the teflon foil over time are regularly calibrated against chemical actinometer experiments

in which $NO_2$ is photolyzed in synthetic air in the chamber (Bohn et al., 2005) . The actinometer experiments were performed at the beginning and end of each intensive phase during JULIAC. A reported accuracy of 18% was achieved and included in the error analysis for the radical budget.

To clarify this, we modify P7 L162 to be 'Reference experiments with clean synthetic air were performed to investigate possible changes in the strength of chamber emissions and to check for instrumental backgrounds. In addition, chemical actinometer experiments were performed in the chamber, in which $NO_2$ was photolyzed in synthetic air, before and after each intensive period. The comparison of actinometric and spectroradiometric $j_{NO2}$ values was used to track and correct for changes in light transmission due to aging of the chamber wall (Bohn et al., 2005).'.

In addition, we change P8 L227 to be 'Photolysis frequencies inside the chamber were derived from solar actinic flux densities measured by a spectroradiometer mounted on the roof of the nearby institute building. Chamber values were calculated using a model approach considering shading effects and the influence of the chamber film (Bohn et al., 2005; Bohn and Zilken, 2005).'

**Comment #5:** P11 L338-339: Was the humidity dependence of k17 accounted for in the calculation of DROx? It seems so from Table 1 where k17 is reported for 1% water but it should be clearly stated in the text.

**Response:** Yes, the water vapour dependence was always considered in the calculations. This is now explicitly stated in the text.

**Comment #6:** P12 L370-371: Please include $O_3$+alkenes in the list of minor Ox destruction pathways

**Response:** Corrected. The modified sentence reads now 'This calculation neglects minor $O_X$ destruction processes such as the reaction of $O_3$ with $NO_2$, OH, $HO_2$, Cl or alkenes since they did not play a notable role during the day in this campaign.'

**Comment #7:** Figures 3, 4, S3, S4: Please indicate in the caption what the error bars represent for OH. Also please add error bars for the other measurements.

**Response:** We add on the captions 'The vertical bars represent $1\sigma$ statistical errors.'

**Comment #8:** Figures 8-12 & S6-S7: Please clarify in the caption whether uncertainties are displayed as 1-sigma? 2 sigma? Other?

**Response:** Corrected.

**Anonymous Referee #2**

**Comment #9:** In the abstract (lines 39-41), the authors state that the missing OH source "consists likely of a combination of a missing primary radical source (0.5 ~ 1.4 ppbv $h^{-1}$) and a missing inter-radical $HO_2$ to OH conversion reaction with a rate of up to 2.5 ppbv $h^{-1}$." However, there appears to be little discussion of this potential OH source/$HO_2$ sink in the paper, except briefly on page 29 (lines 670-671) and page 34 (line 893) and it is not mentioned in the conclusions. If this is a major finding as suggested in the abstract, it should be emphasized more in the manuscript.

Response: The chemical mechanisms behind the inter-radical $HO_2$ to OH conversion independent from NO remains difficult to identify. A potential path driven by halogen oxides is discussed in section 4.2.4 although the conclusion is that the concentration of halogen oxides needed to explain the discrepancy is not realistic for the JULIAC campaign. Although it is mentioned in the conclusion (P37 L994-995), we agree that it was not emphasized as the review's comment. We extended the conclusion in the newer version.

**Comment #10:** The authors state that photolysis frequencies were "were derived from the solar actinic flux densities measured by a spectroradiometer mounted on the roof of the nearby institute building." Given that an underestimation of radical production from photolysis could account for the missing OH radical source, the authors should clarify how potential differences in photolysis rates inside versus outside of the chamber were accounted for in their budget calculations.

**Response:** See our response to Comment #4 by Referee #1 who asked a similar question.

**Comment #11:** The authors measured total OH reactivity and use it to determine the total OH loss rate. However, as illustrated in Figure 5 there appears to be significant missing OH reactivity when compared to the calculated reactivity from measured OH sinks. Unfortunately, there is little discussion about the potential composition of the missing OH reactivity. The paper would benefit from a brief discussion of the missing OH reactivity and whether unmeasured OVOCs may be responsible. While the authors suggest that OVOCs such as acetaldehyde, methyl vinyl ketone, methacrolein, and methylglyoxal do not contribute significantly to radical production, have the authors considered other potential unmeasured OVOCs, perhaps through a model of the chemistry, that may be contributing to the missing reactivity as well as be a potential unmeasured radical source?

**Response:** Modelled OVOCs can often help close the gap between measured and modelled OH reactivity. Model simulation on the $k_{OH}$ were performed in the studies by Kanaya et al. (2012) and by Elshorbany et al. (2012) both analyzing the data collected during the HOxCOMP campaign at the same site as the JULIAC

campaign. The study by Kanaya et al. (2012) tried to match the measured reactivity by increasing primary emissions while the study by Elshorbany et al. (2012) showed an increase of OH reactivity due to oxygenated secondary products (in particular, from isoprene oxidation). Although the measured $k_{OH}$, which was in the range as measured during the JULIAC campaign, could be matched by the modelled one in both studies, similar discrepancies in the OH and $HO_2$ radical budgets remained unless an unknown loss process for $HO_2$ radicals that would recycle OH was introduced. If the missing OH reactivity (~ 2.5 s$^{-1}$) would be all due to glyoxal, an additional OH production of 0.3 ppbv h$^{-1}$ could be expected which would not be enough to explain the missing radical source observed similar to what was found for the HOxCOMP campaign.

We added 'During the HOxCOMP campaign performed in 2005 at the same site as the JULIAC campaign, the modelled OH reactivity could be matched with the measured reactivity by including either additional primary emissions (Kanaya et al., 2012) or model-produced oxygenated secondary products (Elshorbany et al., 2012). Neither of the additional species contributed enough to close the radical budgets. If it is assumed that the missing OH reactivity (2.5 s$^{-1}$) is all due to glyoxal (9 ppb) an additional OH production of 0,3 ppb h$^{-1}$ could be expected. This would still not be enough to close the radical budget suggesting that unmeasured OVOCs do not play a large role.' on P19 L483.

**Comment #12:** P2, L63: The Griffith et al. 2016 reference reports urban measurements. Did the authors mean to cite Griffith et al., Atmos. Chem. Phys., 13, 5403–5423, 2013, which reports measurements in a forest environment?

**Response:** Corrected.

**Comment #13:** In Figure 12 I assume that the numbers at the top of the figure represent the number of points in each NO bin. This should be clarified in the caption. Also, the uncertainty should be clarified.

**Response:** Done.

**References**

Bohn, B., Rohrer, F., Brauers, T., and Wahner, A.: Actinometric measurements of $NO_2$ photolysis frequencies in the atmosphere simulation chamber SAPHIR, Atmos. Chem. Phys., 5, 493-503, doi:10.5194/acp-5-493-2005, 2005.

Bohn, B., and Zilken, H.: Model-aided radiometric determination of photolysis frequencies in a sunlit atmosphere simulation chamber, Atmos. Chem. Phys., 5, 191-206, doi:10.5194/acp-5-191-2005, 2005.

Elshorbany, Y. F., Kleffmann, J., Hofzumahaus, A., Kurtenbach, R., Wiesen, P., Brauers, T., Bohn, B., Dorn, H.-P., Fuchs, H., Holland, F., Rohrer, F., Tillmann, R., Wegener, R., Wahner, A., Kanaya, Y., Yoshino, A., Nishida, S., Kajii, Y., Martinez, M., Kubistin, D., Harder, H., Lelieveld, J., Elste, T., Plass-Dülmer, C., Stange, G., Berresheim, H., and Schurath, U.: HOx budgets during HOxComp: A case study of HOx chemistry under NOx-limited conditions, J. Geophys. Res.: Atmos., 117, doi:10.1029/2011JD017008, 2012.

Fuchs, H., Holland, F., and Hofzumahaus, A.: Measurement of tropospheric $RO_2$ and $HO_2$ radicals by a laser-induced fluorescence instrument, Rev. Sci. Instrum., 79, 084104, doi:10.1063/1.2968712, 2008.

Jenkin, M. E., Valorso, R., Aumont, B., and Rickard, A. R.: Estimation of rate coefficients and branching ratios for reactions of organic peroxy radicals for use in automated mechanism construction, Atmos. Chem. Phys., 19, 7691-7717, doi:10.5194/acp-19-7691-2019, 2019.

Kanaya, Y., Hofzumahaus, A., Dorn, H. P., Brauers, T., Fuchs, H., Holland, F., Rohrer, F., Bohn, B., Tillmann, R., Wegener, R., Wahner, A., Kajii, Y., Miyamoto, K., Nishida, S., Watanabe, K., Yoshino, A., Kubistin, D., Martinez, M., Rudolf, M., Harder, H., Berresheim, H., Elste, T., Plass-Dülmer, C., Stange, G., Kleffmann, J., Elshorbany, Y., and Schurath, U.: Comparisons of observed and modeled OH and $HO_2$ concentrations during the ambient measurement period of the $HO_x$Comp field campaign, Atmos. Chem. Phys., 12, 2567-2585, doi:10.5194/acp-12-2567-2012, 2012.

Nehr, S., Bohn, B., Fuchs, H., Hofzumahaus, A., and Wahner, A.: HO2 formation from the OH + benzene reaction in the presence of O2, Physical Chemistry Chemical Physics, 13, 10699-10708, doi:10.1039/C1CP20334G, 2011.

Nehr, S., Bohn, B., Dorn, H. P., Fuchs, H., Häseler, R., Hofzumahaus, A., Li, X., Rohrer, F., Tillmann, R., and Wahner, A.: Atmospheric photochemistry of aromatic hydrocarbons: OH budgets during SAPHIR chamber experiments, Atmos. Chem. Phys., 14, 6941-6952, doi:10.5194/acp-14-6941-2014, 2014.